# Phylogeny of caucasian rock lizards (*Darevskia*) and other true lizards based on mitogenome analysis: Optimisation of the algorithms and gene selection

**Marine Murtskhvaladze**[1,2], **David Tarkhnishvili**[1] *, **Cort L. Anderson**[1], **Adam Kotorashvili**[2]

1 School of Natural Sciences and Engineering, Ilia State University, Tbilisi, Georgia, 2 L. Sakvarelidze National Center for Disease Control and Public Health, Tbilisi, Georgia

* david.tarkhnishvili@iliauni.edu.ge

**Data Availability Statement:** All relevant data are within the manuscript and its Supporting Information files.

## Abstract

We generated a phylogeny for Caucasian rock lizards (*Darevskia*), and included six other families of true lizards (Lacertini), based on complete mitochondrial genome analysis. Next-generation sequencing (NGS) of genomic DNA was used to obtain 16 new mitogenomes of *Darevskia*. These, along with 35 sequences downloaded from GenBank: genera *Darevskia*, *Zootoca*, *Podarcis*, *Phoenicolacerta*, *Takydromus*, *Lacerta*, and *Eremias*—were used in the analysis. All four analytical methods (Bayesian Inference, BI; Maximum Likelihood, ML; Maximum Parsimony, MP; and Neighbor-Joining, NJ) showed almost congruent intra-generic topologies for *Darevskia* and other lizard genera. However, ML and NJ methods on one side, and BI and MP methods on the other harvested conflicting phylogenies. The ML/NJ topology supports earlier published separation of *Darevskia* into three mitochondrial clades (Murphy, Fu, Macculloch, Darevsky, and Kupinova, 2000), but BI and MP topologies support that the basal branching occurred between *D. parvula* from the western Lesser Caucasus and the rest of *Darevskia*. All topologies altered the phylogenetic position of some individual species, including *D. daghestanica*, *D. derjugini*, and *D. chlorogaster*. Reanalysis after excluding four saturated genes from the data set, and excluding genus *Eremias* gives fully convergent topologies. The most basal branching for true lizards was between Far Eastern *Takydromus* and the Western Eurasian genera (BI). Comparing phylogenetic performance of individual genes relative to whole mitogenome data, concatenated 16S RNA (the least saturated gene in our analyses) and Cytochrome b genes generate a robust phylogeny that is fully congruent with that based on the complete mitogenome.

## 1. Introduction

During the last three decades, full or partial sequences of mitochondrial DNA (mtDNA) have been used for phylogenetics, comparative and evolutionary genomics, population genetics,

**Funding:** MM received scholarship (award #
PS00215196) from Fulbright Scholar Program
https://www.cies.org/; DT grant (award # 217478)
from Shota Rustaveli National Science Foundation
https://www.rustaveli.org.ge The funders had no
role in study design, data collection and analysis,
decision to publish, or preparation of the
manuscript.

**Competing interests:** The authors have declared
that no competing interests exist.

and investigation of the molecular evolution of many species. The reason for such popularity is its high copy number per cell, availability of universal primer sequences, the lack of recombination, and an accelerated mutation rate compared to those of the nuclear DNA [1–7].

More recent phylogenetic studies typically contain more data, from longer DNA sequences and more loci. This in turn has led to revision of previous phylogenies, usually with increased statistical support of the findings, higher resolution, and therefore improved understanding of the evolutionary relationships. This is certainly the case within the Tetrapoda [8–12]. Notwithstanding these revisions, mitochondrial genes are still commonly used because of their obvious advantages, although phylogenies based on the analysis of full mitochondrial genomes have generally proven more robust and less dependent on the particular algorithm used for analysis [6, 13, 14]. The advantages deriving from analysis of full mitogenomes is twofold– 1) there are simply more informative substitutions when comparing individual genes, and 2) this avoids the risk of sequencing nuclear copies, which can be substantial for individual mitochondrial genes [15–17].Saturation of individual genes may produce topologies with a limited resolution, the problem identified in pioneering research using DNA sequences [18], and increasing the length of the sequences helps to overtake this problem.

One potential shortcoming of mitochondrial phylogenies is that this is a single non-recombinant locus, and due to introgression patterns and/or incomplete lineage sorting, the boundaries of species as cohesive genetic systems may not coincide with the boundaries between the mitochondrial haplogroups (e.g. [3, 16, 19, 20]). On the other hand, given the possible incongruence of phylogenies based on the unlinked genes in species with incomplete genealogical concordance, the interpretation of mitochondrial phylogenies is straightforward: it tracks the sequence of bifurcations of the maternal lineages.

Currently, seven thousand complete or nearly complete animal mitogenomes are available from the existing databases, and this number is increasing daily [20]. In this paper, we use mitochondrial genomes of Caucasian rock lizards (genus *Darevskia*; [21]) for reconstructing a species-level phylogeny of this taxon and its position within true lizards of subgenus Lacertinae. We not only specified position of individual taxa within *Darevskia*, but also investigated whether an optimized subset of mitochondrial DNA loci can produce phylogenetic trees with resolution comparable to the complete mitochondrial genome. The latter objective is important because obtaining full mitochondrial DNA sequences is still associated with substantial expense, irrespective of whether conventional sequencing methods or NGS methods are applied; PCR-based sequencing of individual genes remains a widely used alternative for many laboratories without access to NGS technology.

The genus *Darevskia* is peculiar for having a relatively small geographic distribution, but is simultaneously highly speciose. 27 bisexual and 7 parthenogenetic species have been described for this genus, mostly within the Caucasus biodiversity hotspot [22, 23], unusually high species diversity for a non-tropical vertebrate group with such a restricted range. A number of evolutionary studies, covering phylogeny, phylogeography, speciation, and evolution of unisexual reproduction, use *Darevskia* as system for case studies [24, 17, 21, 23–30]. Murphy et al. (1) suggested that the genus has three well-supported clades: *rudis*, *saxicola*, and *caucasica*, named after a representative species from each clade. They inferred phylogeny of 13 sexually breeding rock lizards, but omitted the species from the southern Caspian region. Ahmadzadeh et al. [28] revised this phylogeny, adding species from the southern Caspian region to the analysis, and showed that all of these belong to the *caucasica* clade. Pyron et al. [10]; Roquet et al. [31], and Zheng et al. [11] re-analysed available genetic data for *Darevskia* and included them in their phylogenies of world-wide or western Eurasian squamates. These successive studies showed discrepancies from the phylogeny inferred by Murphy et al. [24]; a number of clades within the existing trees remained disputed. Phylogeny of Lacertidae recently published by

Garcia-Porta et al. [32] is similar to that of [24], and [28] but suggests the most basal branching was between *D. parvula*, which previously was included in clade *D. rudis*, and the rest of *Darevskia* (see also re-analysis of published sequences [23]. Meanwhile, though over 20 mitogenomes of Lacertini have been published during the past decade, the phylogenetic relationships, even among the most extensively studied genera, remain unresolved, due to low statistical support of the respective trees. This uncertainty also includes the position of the genus *Darevskia* [11, 31, 33, 34].

The second important aim of this study is identifying the most phylogenetically informative mitochondrial genes, which could simplify future phylogenetic and phylogeographic studies of true lizards, especially for those researchers who still use PCR-based methods. Increasing the number and size of sequenced fragments asymptotically improves confidence in phylogenetic outcomes, and at some threshold, the results based on different fragments will converge [7]. However, incongruence among different mitochondrial regions may cause contradictions between phylogenies [6]. These authors recommend using complete mitochondrial genomes for inferring reliable phylogenies, rather than sampling individual genes. Identification of the most informative genes should go a long way towards producing phylogenies converging on those based on complete mitogenomes.

We generated complete mitochondrial sequences for 16 *Darevskia* species, and sequence of *D. unisexualis* was downloaded from GenBank (Acc # KX644918 [35]). These species' represent all three clades outlined by Murphy et al. [24]. We omitted a few close relatives, e.g., *D. alpina* and *D. saxicola* which show minute differences from *D. brauneri*, which is included in the analysis [24, 30]; of the southern Caspian species group [28], only on taxon is included, *D. chlorogaster*. Almost all included species are rock-dwelling forms, except for the ground-dwelling *D. derjugini* and *D. praticola*. Our data set fully represent the geography and ecology of the genus, as it includes species from both the Greater and Lesser Caucasus, taxa from the southern Caspian coast, and species that depend on the mesophilic climate of the Western Caucasus and southern Caspian area as well as those from the dry habitats south of the Lesser Caucasus. The data set includes 17 taxa of the genus, three of which are parthenogenetic forms: *D. armeniaca* and *D. dahli* are descended matrilineally from *D. mixta*, while *D. unisexualis* is descended from *D. raddei* [24]. Mitogenomes of true lizard genera *Podarcis*, *Zootoca*, *Phoenicolacerta*, *Takydromus*, *Lacerta*, and *Eremias* were downloaded from GenBank and included in the analyzed dataset together with *Darevskia* species. The analysis reveals a number of novel details in the evolution of *Darevskia* and some of their closest relatives.

## 2. Materials and methods

### 2.1. Sample collection and DNA extraction

Tissue samples (tail tips) were obtained from live individuals collected from 15 locations in Georgia and one in Azerbaijan. The individuals represent 14 bisexual species and two parthenogenetic forms of *Darevskia* (Fig 1). The tail-tips were preserved in 95% ethanol. DNA was extracted from the samples using Qiagen DNeasy Blood and Tissue Kit, according to the manufacturer's instructions (DNeasy Blood & Tissue Handbook 07/2006, [36]). To check for contamination, a negative control (reagents only) was included during extractions, and tested for amplification during PCR.

### 2.2. Shotgun sequencing using Illumina MiSeq platform

Purified DNAs were quantified using 2100 Bioanalyzer Laptop Bundle (Agilent Technologies, Santa Clara, USA), and Qubit$^{\circledR}$ 2.0 fluorometer (Invitrogen, Grand Island, USA). High quality DNA was sonicated to an average size of 500 bp. TrueSeq DNA kit v3.1 was used for library

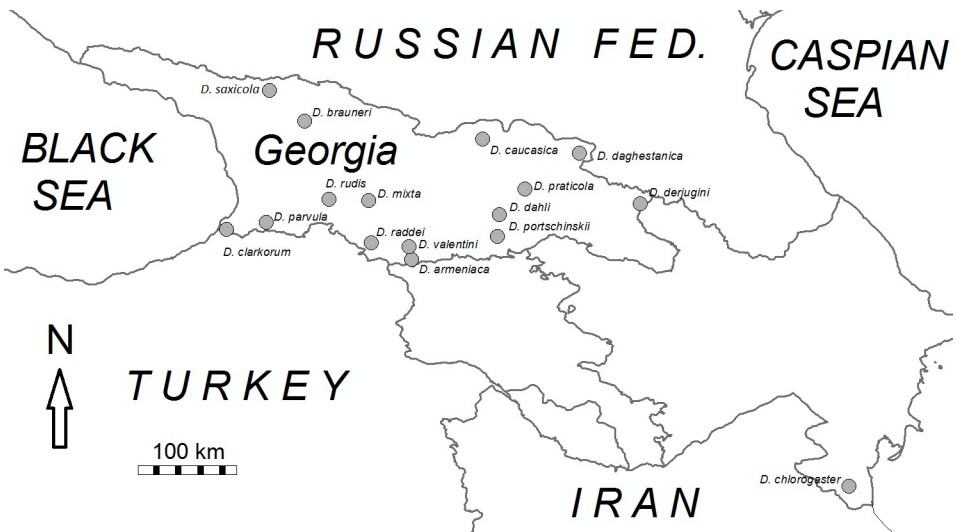

**Fig 1. Geographic location of the samples used in this paper.** Grey dots indicate the sampling locations.

preparation, following manufacturer's recommended protocol. Adaptor-ligated DNA molecules were enriched by 10 cycles of PCR and amplified library purified by AMPure XP beads (Beckman Coulter cat# A63881), followed by elution with Resuspension Buffer. Concentration of the adapter-ligated DNA was calculated using the following formula: x nM = (Concentration ng/µl * x 10000000)/(660g/mole) (bp Avg. Length) to determine the volume of each sample to include in pooled DNAs, and were diluted to 2 nM final concentration of library mixtures. For DNA denaturation, 2N NaOH was added to equal volumes of the normalized DNA libraries. The DNA libraries were diluted to 20 pM concentration in chilled HT1 buffer. For internal control, 1% PHiX DNA was supplemented into the 20 pM denatured DNA solution, to observe the efficiency of DNA incorporation during DNA sequencing. Libraries were loaded into the cartridge, and sequenced as multiplex two-read libraries for 600 cycles. Fluorescent images were analysed according the Illumina base-calling pipeline 1.4.0 for obtaining FASTQ-formatted sequence data.

## 2.3. Genome assembly

We performed quality control analysis using FastQC v. 0.11.5 software [37]. NGS data trimming and adapter clipping was performed with Trimmomatic version 0.33 [38]. Quality trimmed reads from all species were mapped to *Darevskia unisexualis* mitogenome (GenBank accession # KX644918, [35]). CLC-Bio Genomics Workbench v 8.0 (www.qiagen.com/us/search/clc-genomics-workbench) was used for mapping reads. Parameters for assembling reads were as follows: mismatch cost 2; deletion cost 3; insertion cost 3; length fraction 0.5; similarity fraction 0.7; Consensus sequences were exported with at least 10X coverage. If the coverage did not meet this threshold, N (ambiguity) was inserted. At positions with disagreements in base calls between reads (minimum 10%), an appropriate ambiguous nucleotide symbol was inserted.

Sequence alignment was performed with Geneious R11 (https://www.geneious.com) software. Within *Darevskia* sequences, all gene annotations and boundaries of each discrete segment of mt DNA were determined by sequence comparison with their counterparts in the published mitogenome of *Darevskia unisexualis* [35].

## 2.4. Phylogenetic analysis

Complete mitochondrial genomes of the following species: *Darevskia parvula*, *D. rudis*, *D. portschinkii*, *D. valentini*, *D. praticola*, *D. brouneri*, *D. saxicola*, *D. raddei*, *D. chlorogaster D. dagestanica*, *D. derjugini*, *D. caucasica*, *D. clarcorum*, *D. mixta*, *D. dahli*, and *D. armeniaca* were assembled and annotated. The complete mitogenome of *D. unisexualis* was downloaded from GenBank (accession # KX644918, [35]). Mitogenomes of other species included in the analysis were: *Podarcis muralis* (Acc # NC011607, FJ460597; [15]), *P. siculus* (Acc # NC011609, FJ460598; [15]), *Lacerta viridis* (Acc # NC008328 [39]) *L. agilis* (Acc # NC021766, KC990830; Qin & Tao, unpublished), *L. bilineata* (Acc # KT722705, NC028440; [40], *Takydromus tachydromoides* (Acc # NC008773 [41], *T. amurensis* (Acc # NC030209, KU641018; [42]) *T. sexlineatus* (Acc # NC022703, KF425529 [43]), *T. wolteri* (Acc # NC018777, JX181764; [44]), *Zootoca vivipara* (Acc # NC026867 [45]) and *Phoenicolacerta kulzeri* (Acc # FJ460596 (16)), *Eremias argus* (Acc # IQ086345, NC016755; Kim et al., unpublished), *E. multiocellata* (Acc # KJ664798, KM257724; NC025304 [46]), *E. przewalskii* (Acc # KM507330, NC025929; [47]), *E. stummeri* (Acc # NC029878, KT372881; [48]) *E. vermiculata* (Acc # NC025320, KP981389, KM104865; [46]) (Acc # KP981388; [46]), and *E. velox* (Acc # KM359148; [48] *E. brenchleyi* (Acc # EF490071; NC011764 Rui et al., unpublished).

For comparison of the phylogenetic performance of individual parts of the mitogenome, we separately reconstructed phylogenies of the following individual genes or groups of genes: 1) the thirteen mitochondrial protein coding genes, 2) concatenated tRNAs and 3) ribosomal (12S and 16S) RNA genes. In squamate species' control region, mtDNA has heterogeneous base composition and long insertions and therefore shows biased phylogenetic signal [39, 47, 49, 50], hence we excluded these segments from our data set. For determining the best substitution models, we analysed each protein coding gene for 1st, 2nd and 3rd codon positions with PartitionFinder v.2. [51]. The parameters for each gene partition were as follows: branch length = linked; models = all; model selection = BIC; schemes = all.

The phylogeny based upon the complete mitogenome, as well as the trees based on the analysis of individual genes for partitioned mtDNA sequence, were estimated with Maximum Likelihood (ML) and Bayesian Inference (BI). ML analysis was performed using MEGA ver. 7.0 [52]. The respective models and prior specifications for ML analysis were set according to the models of nucleotide evolution shown in Table 1, and statistical support for branching patterns was estimated by 500 bootstrap replications. BI was performed using BEAST v.2.4.5 [53] and BayesPhylogenies v.1.1 [54]. We built the input file with evolutionary models, tree priors and Markov Chain Monte Carlo (MCMC) options using the BEAUTi utility included in the BEAST package, without the assumption of constant evolutionary rates. We used Relaxed Uncorrelated Lognormal Clock set for all genes, and Yule process of speciation as a tree prior (it requires only one sequence per species). For mitogenome phylogenies, BEAST was run with 200 million generations, sampling every 5,000 generations, and 27–60 million generations for each gene. We used Tracer v 1.6 [55] to check the runs for convergence (burn-in = 10%) and to ensure that all effective sample size parameters (ESS) were higher than 200, as recommended in the manual. Runs were combined with LogCombiner, and afterwards TreeAnnotator (both included in the BEAST package) was used to summarize the trees in a 50% majority rule consensus tree representing the posterior probability distribution. We used the output of the same analysis using BEAST for both reconstructing phylogeny and inferring dates of split between the clades within Lacertini and within the genus *Darevskia* (Fig 6). For calibration of the tree, we used fossil analysis-based timescale developed by [32], setting the average branching time within Lacertii 37 mya.

**Table 1. Nucleotide mutation models, number of informative sites\*.**

| Gene | bp | BM | Tr/Tv corr | VSD | ML-U | BI-U |
|------|------|------|-----------|------|------|------|
| 12S | 959 | GTR+G+T | 0.84 | 161 | 8 | 3 |
| 16S | 1554 | GTR+G+T | 0.93 | 306 | 1 | 1 |
| ATP8 | 161 | TN93+G+I | 0.44 | 74 | 6 | 4 |
| ATP6 | 681 | HKY+G+I | 0.80 | 239 | 9 | 5 |
| COX1 | 1544 | GTR+G+T | 0.70 | 430 | 4 | 3 |
| COX2 | 687 | TN93+G+I | 0.74 | 204 | 6 | 2 |
| COX3 | 783 | GTR+G+T | 0.66 | 241 | 6 | 1 |
| NADH1 | 968 | TN93+G+I | 0.65 | 325 | 4 | 3 |
| NADH2 | 1032 | TN93+G+I | 0.61 | 409 | 2 | 3 |
| NADH3 | 345 | HKY+G+I | 0.62 | 126 | 9 | 2 |
| NADH4L | 296 | HKY+G+I | 0.71 | 124 | 5 | 3 |
| NADH4 | 1380 | HKY+G+I | 0.74 | 541 | 2 | 1 |
| NADH5 | 1823 | TN93+G+I | 0.79 | 611 | 5 | 2 |
| NADH6 | 515 | TN93+G+I | 0.70 | 177 | 6 | 2 |
| Cyt B | 1140 | GTR+G+T | 0.68 | 390 | 1 | 1 |
| MG\*\* | 15479 | GTR+G+T | | 4811 | | |

\*bp—base pair number. BM—Best nucleotide substitution models (under BIC). VSD—Variable sites within *Darevskia* genus. VSN—Variable sites within lacertini subfamily. ML-U—The nodes within the topology of genus *Darevskia* that are unresolved or non-coinciding with those from mitogenome analysis based on the ML method. BI-U—The nodes within the topology of genus *Darevskia* that are unresolved or non-coinciding with those from mitogenome analysis based on the BI method. \*\* Mitogenome with the control region excluded

Two additional methods were applied for estimating the consistency of inferred phylogenies: maximum parsimony (MP), and the distance-based Neighbor-Joining (NJ) tree reconstruction. MP analysis was implemented using PAUP 4.0 [56], with all nucleotide changes at all positions weighted equally. Heuristic searches were run using random addition of taxa, tree bisection and reconnection algorithm (TBR). A distance-based NJ analysis was implemented using MEGA ver. 7.0 [50] by estimating the models listed in Table 1. Statistical support for branching patterns was estimated by 500 bootstrap replications for both MP and NJ analyses. All trees were visualized in FigTree v1.4.3 (available at http://tree.bio.ed.ac.uk/software/figtree/).

Because the ML and BI analyses did not show fully consistent phylogenetic patterns (see the results section), we repeated the analysis after removal of genes that showed the highest saturation of transitions for ingroup taxa. The latter ones were identified running the software DAMBE [57], as recommended by [58]. For individual mitochondrial genes we calculated correlation coefficients between the transition and transversion rates, assuming that the saturation of transitions will decrease the correlation.

We compared the deviations of topologies (conflicting and unresolved nodes) based on the individual genes from the topology based on the complete mitochondrial genome built with BI and ML for: The sum of the conflicting, coinciding, and unsupported nodes was considered a measure of similarity between the compared trees, along with the saturation index (correlation between the transition and transversion rates).

## 3. Results

### Genome organisation

Sequences of complete mitochondrial DNAs of *Darevskia* species were deposited in GenBank with accession numbers MH481130-37; MG704915-21. In all the species we have > 20X

coverage. The length of mitogenomes are different in *Darevskia* species' and varies from 16301–20478 bp because of 890 bp—2.5 kb insertions in the noncoding D-loop region. The structural arrangement of the 13 protein coding genes, rRNA genes, 22 tRNA genes, and control region corresponds to the typical vertebrate pattern [2, 59–61]. Some of the protein coding genes (*ATP6* and *COX 3*; *NADH 4L* and *NADH 4*; *NADH 5* and *NADH 6*) showed partially overlapping sequences, we also found spacing sequences up to 50 bp within the species between the different coding features (*NADH 1* and *tRNA Ile*; *tRNA Asn* and *tRNA Cys*; *tRNA Glu* and Cytochrome B). The average A+T content is 59.4% in all *Darevskia* species, other vertebrate species show similar composition [15, 35, 41, 59]. Within the studied species for protein coding genes, ATG is commonly the start codon, except *COX1*, which starts with TGT. In addition, *NADH 5* starts with ACC in species: *D. mixta*, *D.dahli*, *D. armeniaca*, *D. caucasica* and *D. clarkorum*, with ATC in *D. chlorogaster* and *D. derjugini*, GTG in *D. parvula* and ACA in *D. praticola*. AGG is the stop codon for the *COX 1* gene, and TAA for *Cyt b*, *ATP 6*, *ATP 8*, *NADH 4L* and *NADH 5*; the observed lengths of the protein coding genes were similar to those found in other reptile mitochondrial genomes [38, 40, 41, 62].

## Phylogenies based on complete mitochondrial genome

The topologies of the reconstructed phylogenetic trees based on ML and NJ methods had high (>96) bootstrap support for 43 out of 48 nodes and moderate (>60) bootstrap support for the other 4 nodes. The tree based on BI had posterior probabilities close to 1 for all 48 nodes. The tree inferred with Maximum Parsimony had bootstrap support < 50 for the majority of nodes linking the genera included in the analysis. The topologies based on ML and NJ methods on the one hand, and BI and MP methods on the other hand, were noncongruent in the parts describing phylogenetic relations among genera, and the position of *D. parvula* within the genus *Darevskia* (Fig 2A and 2B).

Saturation analysis using DAMBE software conducted for individual genes (S1 Fig) showed the highest transition saturation level for mitochondrial genes *ATP8*, *NADH2*, *NADH3*, and *NADH1*. These genes showed the lowest correlation coefficients between the transition and transversion rates: 0.44, 0.61, 0.62, and 0.65, respectively. The saturation curves also indicate that saturation of transitions is skewed by the presence of *Eremias* species (the taxon genetically most distant from the other analyzed species) in the data set (Fig 3). For this reason, we repeated the analyses after excluding *Eremias* from the species set and removing the four abovementioned genes from the dataset. After pruning the dataset, the resultant trees inferred with BI and MP methods did not change, and all four methodologies showed fully congruent strict consensus trees.

This consensus phylogeny of *Darevskia* showed differences from previously published topologies. Specifically, the basal branching occurred between *D. parvula* and the rest of *Darevskia* and not between '*D. rudis*' clade and the species that belong to the clades '*D. caucasica*' and '*D. saxicola*'. For the clade comprised of the species from the Greater and the Lesser Caucasus (excluding the southern Caspian area), the basal branching was between *D. daghestanica* and *D. derjugini* + *D. caucasica* + *D. mixta* + *D. clarkorum*, and *D. daghestanica* is not a sister taxon of *D. caucasica* as earlier researchers suggested; the basal branching for the clade *D. rudis* + *D. valentini* + *D. portschinskii* is between *D. rudis* and the two other species and not between *D. portschinskii* and *D. rudis* + *D. valentini*; the most basal branching in the clade "*D. caucasica*", *sensu* Murphy et al. (1), is between *D. raddei* and the rest of the species and not between those and the southern Caspian *D. chlorogaster*, in contrast with findings of Ahmadzadeh et al. (28) (Fig 4). From the other lizard species, all analyses suggested the most genetically distant position of *Takydromus* relative to other Lacertini, and closer placeent of *Podarcis*

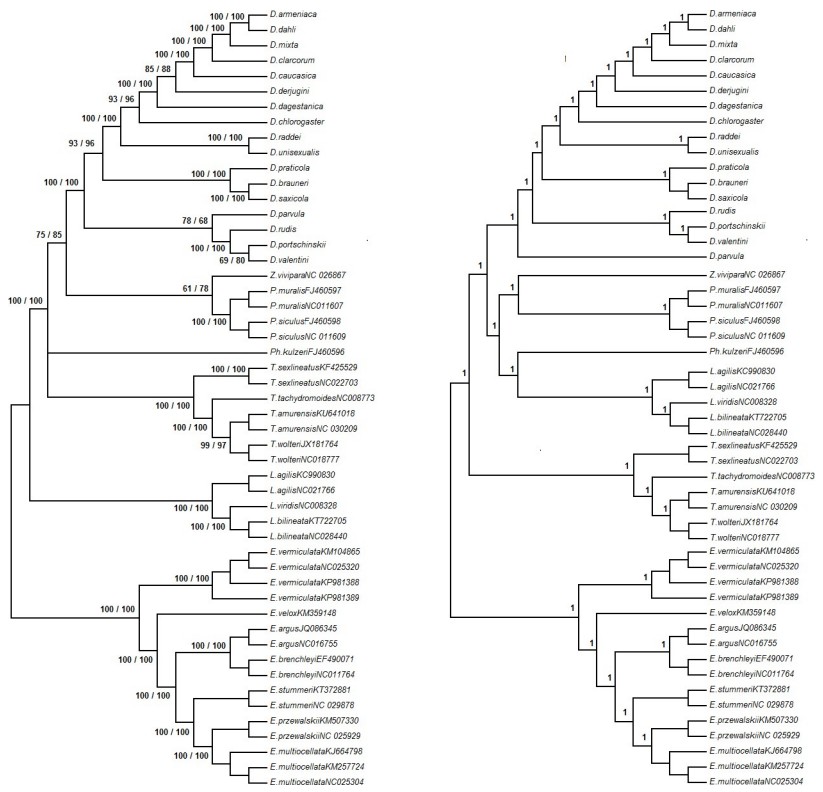

**Fig 2. Tree topologies, based on the complete mitochondrial genome of *Darevskia* and selected species of Lacertini.** A) The topologies inferred from ML and NJ analyses. Bootstrap supports for ML/NJ analyses left to the nodes. B) The topology based on BI analysis. Posterior probabilities left to the nodes. The nodes with bootstrap support below 51 or posterior probabilities below 0.51 shown as unresolved nodes/ polytomies. The topology based on MP analysis showed multiple polytomies but the nodes with bootstrap support exceeding 51 were not conflicting with the tree inferred using BI analysis (results not shown).

and *Zootoca*, relative to *Phoenicolacerta* and *Lacerta*; all analyses showed coinciding topologies within *Takydromus* and *Lacerta*.

## Phylogenies based on individual mitochondrial genes and their combinations

Relevant mutation models of each mitochondrial gene, the number of variable sites, non-coinciding and polytomic nodes are shown in Table 1. Different gene trees suggest different levels of deviation from the mitogenome-based topologies. The most informative gene, providing fewest deviations from the full mitogenome-based topology, and convergent for ML and BI methods, was 16S (1554 bp). This gene and also Cyt-B mitochondrial gene produced only one node non-coinciding with that produced by the analysis of complete mitogenome. 16S also showed the highest (0.93) correlation between transitions and transversions (saturation index), i.e. was least saturated. ATP8 gene, conversely, showed both the highest saturation index and the most conflicting and unresolved nodes. However, it was no overall correlation between the saturation index and the number of unresolved nodes. COX3 (783 bp), NADH2 (1032 bp), NADH4L (294 bp), and Cytochrome b (1140 bp) showed no conflicting positions with mitogenome-based topology when ML method was applied (and multiple conflicting positions if BI was used), but all except Cytochrome b showed multiple unresolved taxa. The latter gene showed only one unresolved group when ML was used. Finally, BI and ML analysis

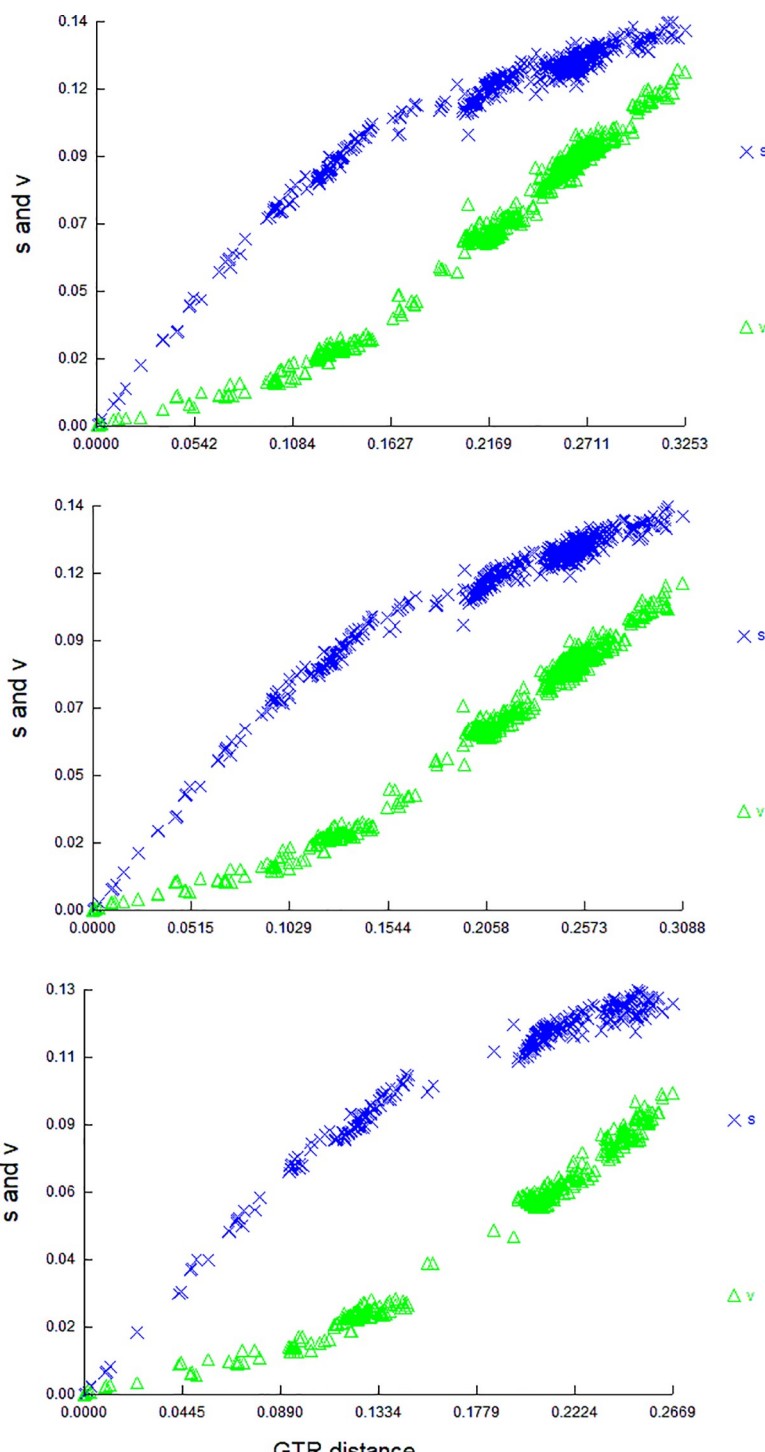

**Fig 3. Saturation curves for transitions and transversions.** Upper panel—the distances among both ingroup and outgroup taxa, full mitogenome without control region; middle panel—the distances among the ingroup and outgroup taxa, full mitogenome without control region and four the most saturated genes: ATP8, ND2, ND3, and ND1. Lower panel—the distances among Lacertini (*Eremias excluded*), full mitogenome without control region and four the most saturated genes.

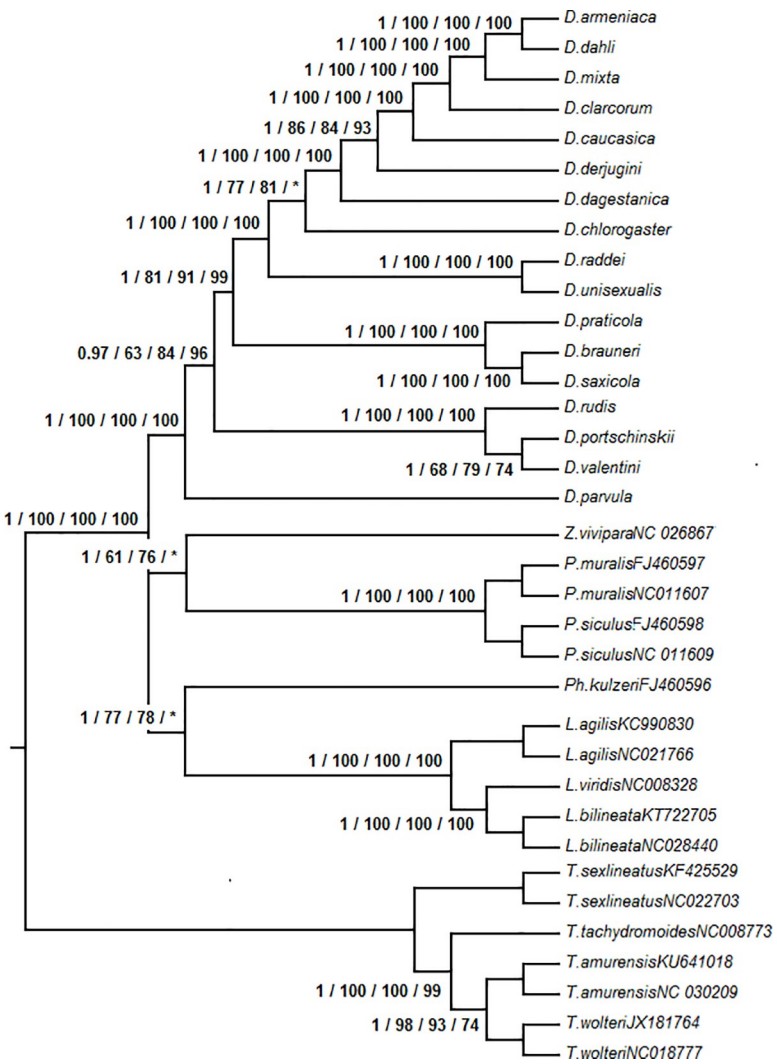

**Fig 4. Strict consensus phylogenetic tree of Lacertini, based on the four applied methods, rooted at midpoint.** The tree is based on the complete mitochondrial genome with the excluded control region and genes ATP8, ND2, ND3, and ND1. Posterior probabilities (BI) and bootstrap supports (ML / NJ / MP) left to the nodes. *—bootstrap support below 50. The politomy reflects the disagreement between the BI and MP, on one side, and ML and NJ methods.

of the concatenated genes *16S+Cytb* showed all nodes coincident, and produced a topology of the ingroup taxa fully congruent with that derived from complete mitochondrial genome sequences using the same analytical methods. In general, the least informative genes showed the highest saturation index (low correlation between the transitions and transversions), and the most informative genes exhibited the lowest saturation index (Table 1). However, negative correlation between the saturation index and deviation from the full mitogenome was not significant (Rxy = -0.32, P>0.05).

## 4. Discussion

The tree topology, based on the analysis of full mitochondrial genome became congruent for the described inference methods only after removal of the most saturated sequences and the most distant lacertid genus (*Eremias*) from the mitogenomic analysis. The resulting

mitochondrial phylogeny has strong support for both the relationships within *Darevskia* and among the included genera of true lizards. It indicates that the most basal branching within the genus *Darevskia* was between *D. parvula* and the rest of the species. The topology based on the mitochondrial genome is substantially different from recent published phylogenies based on meta-analysis for a majority of described squamate species [10, 11] but almost congruent with that presented in [32], based on the analysis of selected mitochondrial and nuclear genes. The discrepancies from the earlier published phylogenies of *Darevskia* [24, 23, 28] are also not so stark.

## 4.1. Differences from earlier findings

Since molecular data have become commonplace, a number of researchers have tried to use data of this kind to resolve relationships of lacertid lizards, including the genus *Darevskia*. The most significant recent papers on the lacertid family (including *Darevskia*) phylogeny are those of [26, 33, 63–66], but also more inclusive papers of [10, 11, 31], and in particular the paper of Garcia-Porta et al. [32] presenting exhaustive phylogeny of true lizards.

The consensus topology presented in this paper is congruent with some earlier publications, both those describing *Darevskia*, and the position of other Lacertid genera included in our analysis. The basal branching between the eastern Asian *Takydromus* and the western Eurasian Lacertini supports earlier findings of Harris [63], although our analuysis contradicts the results of [10, 31, 32] and [64]. Sister status of genera *Podarcis* and *Zootoca* is consistent with the results of [33] and [64], although contradictory to [10] and [32].

The earlier published phylogenies of *Darevskia* are less controversial. [24] inferred the mitochondrial phylogeny based on analysis of a 2851 bp fragment using MP, and this phylogeny has undergone only minor corrections in subsequent analyses, including the present paper (Fig 5) They studied 13 bisexual lizards: *D. rudis*, *D. valentini*, *D. portschinskii*, *D. parvula*, *D. saxicola*, *D. alpina*, *D. praticola*, *D. raddei*, *D. derjugini*, *D. caucasica*, *D. daghestanica*, *D. mixta*, and *D. clarkorum*. We did not include *D. alpina* in our analysis, however, this species is matrilineally very close to *D. saxicola* [24, 23]. Instead, we have in our analysis *D. chlorogaster*, a representative of an important Southern Caspian clade.

The position of *D. parvula* is historically subject to debate. Murphy et al. [24] placed this species as a sister lineage of the clade *D. rudis* + *D. valentini* + *D. portschinskii*, the conclusion supported by sequences analysis of a bigger species set of *Darevskia* [28]. Earlier, Grechko et al. [67], based on the restriction endonuclease analysis of highly repetitive DNA, hypothesized a basal branching between *D. parvula* and the rest of *Darevskia*; they supported their views based on satellite DNA analysis [68]. Tarkhnishvili [23] re-analyzed published data with Bayesian algorithm, which supported this topology. Finally, very inclusive analysis of Garcia-Porta et al. [32] confirmed supported this view. We suggest that the full mitogenome analysis presented here provides additional evidence of basal branching between *D. parvula* and other *Darevskia*.

The topologies of [24, 28, 32] (Fig 5B and 5C) have a few discrepancies when compared to our results: (1) all suggest *D. rudis* and not *D. portschinskii* to be a sister species for *D. valentini*; (2) all suggest sister status of *D. caucasica* and *D. daghestanica* and (3) the two former papers suggest a closer relationship between *D. mixta* and *D. derjugini* rather than with *D. caucasica*, whereas the results of Garcia-Porta et al. [32] coincide with the ours. In addition, [28] suggests the basal branching between *D. chlorogaster* (and not *D. raddei*) and the other species of the clade *D. "caucasica"*, and [32] suggest sister status of *D. chlorogaster* group with *D. raddei*. Simultaneously, all methods applied to complete mitochondrial genome suggest that the second most basal branching within this clade is between *D. chlorogaster* and the rest of the

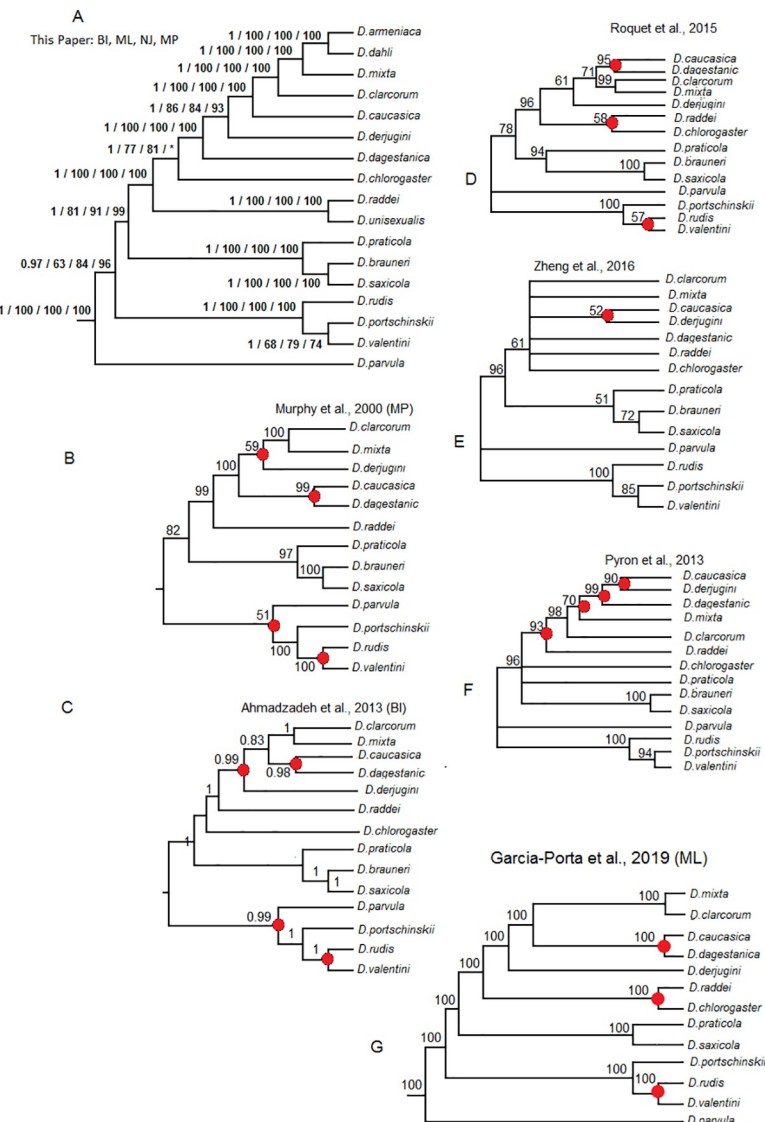

**Fig 5. Topologies of *Darevskia*, inferred by different authors and that from the present paper.** (A) topology based on the complete mitochondrial genome, inferred with ML, NJ, MP, and BI methods (this paper); (B, C) topologies based on the cytochrome b mt-DNA gene fragments (MP, (1); BI, (27); (D, E, F) topologies taken from the metadata-based trees of squamates of (11), (61) and (30); (G) topology based on the combination of mitochondrial and nuclear genes, ML method [32]. Only position of taxa also used in this paper is shown. Threshold level for polytomies is bootstrap value 50 or posterior probability 0.5. Green nodes are those coinciding with our ML tree.

species. These differences from our topologies can be explained with insufficient phylogenetic signal of the DNA sequences analyzed., Simultaneously, some sequences might be derived from pseudogenes, the existence of which has been proven for few species of *Darevskia* [15, 17].

Recent papers based on meta-analysis [10, 11, 31] showed topologies very different from both earlier authors that worked with mitochondrial DNA of *Darevskia*, and our data. The tree of Roquet et al. [31] is concordant with that of Murphy et al. [24] and [28] but *D. raddei* is clustered with *D. chlorogaster* and not with the western Caucasus representatives of the clade *D. "caucasica"*, similar to [32] (Fig 5F). [11] inferred the same result as [28] and [31] on the

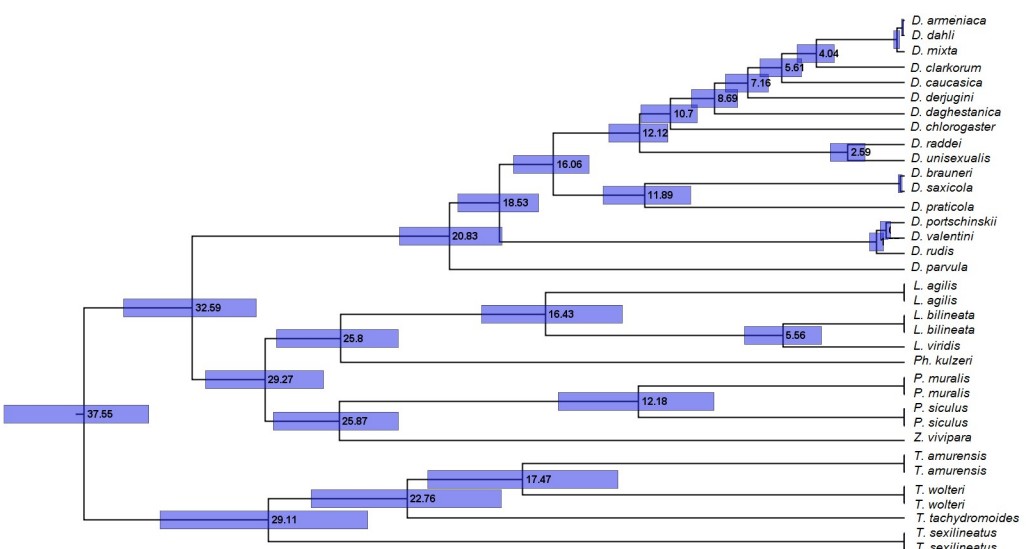

**Fig 6. Time tree inferred with Bayesian model (software BEAST) with the root of Lacertinae subfamily placed after [32].** The numbers at nodes–ages in millions of years; horizontal bars– 95% confidence interval.

position of *D. chlorogaster*, and placed *D. derjugini* in the same clade with *D. caucasica* + *D. daghestanica* and not with *D. mixta* + *D. clarkorum* (Fig 5D). [10] showed topology very different from both published papers and inferred in this study (Fig 5E).

## 4.2. Geography, adaptation, and differentiation of *Darevskia*

Different researchers evaluate the time of separation of *Darevskia* from its closest relatives between 10–68 mya [69, 70]. Most authors, including [10, 11, 71, 72] suggest divergence occurred between 40–45 mya. Garcia-Porta et al. [32] estimated the split time between *Darevskia* and its closest relative *Iranolacerta* as 26 mya; the same authors dated diversification of subfamily Lacertinae to 37 mya. The time of divergence between the most distant lineages of *Darevskia* is estimated to be between 16.8 mya (72) and 46 mya (70), with modal values between 20–30 mya [10, 11, 31, 71], but quite precise estimate of [32] based on the fossil dating is 15.13 mya. The rates of reptile mitogenome evolution vary depending on the taxonomic group and even body size of animals [73]. For this reason we refrain from conclusions on the exact calibration of a molecular clock, however, relying on that of [32] more than the other estimates. Below, we discuss an important geological landmark that might have triggered divergence of the four basal branching events within the genus.

Such a landmark is the Middle Miocene climate transition (MMCT; [74]), ca. 18 mya. Since that time, temperature and humidity on Earth have fallen, and the amplitude of fluctuations has increased, all this accompanied with landscape transition and fluctuations of sea level. At that time, the current Caucasus Isthmus was covered by the ancient Paratetis Sea, and habitats were separated between the Anatolian landmass, the Iranian landmass, and the Caucasus Island, in place of the current Greater Caucasus Mountains [75, 76]. Temporary junctions among these land masses might have occured due to humidity decline after MMCT, permitting dispersal of lizard populations followed by geographic isolation after a rise in sea level. If the root of the tree of subgenus Lacertinae is set, after Garcia-Porta et al. [32] to 37.55 mya, then the time of first split among the lineages of *Darevskia* varies between 18–23 mya (95% HPD; Fig 6), which is in line with MMCT. The separation of *D. saxicola* clade, then, should

occur 14.5–18 mya, which could be associated with the first temporary contacts between the Caucasus Island and Anatolian Mainland (the most of the representatives of this clade are currently found exclusively in the Western Greater Caucasus, from where other lineages of *Darevskia* are absent). The *caucasica* clade (isolated 10.5–13.5 mya; Fig 6) was probably associated with the Iranian plateau (where a substantial part of the species that belong to this clade are still found–[28]. The first junction between this land and Anatolia is dated to 13–14 mya [75].

D. parvula, which is the most evolutionary distant from the rest of *Darevskia* species, is found throughout the western Lesser Caucasus, where it coexists with the representatives of the next basal clade of *Darevskia*. [68] suggest that the range of *D. parvula* is likely a region of differentiation for *Darevskia*. Hence, the northeast of the Anatolian Landmass, which is associated with the current Lesser Caucasus, is a likely area where the ancestral lineages of *Darevskia* originated (Fig 6). Under this scenario, the dating of [11] or [72]–overestimates the time of diversification between *Darevskia* lineages, and that of [32] should be considered as the most accurate one.

## 4.3. Substitution of full genome analysis with sequencing of individual genes

The lesson learned from this study is that even nodes with high statistical support may not coincide for different methods of tree building, and the tree building algorithm appears to be an important determining factor for topology. Convergence of topologies obtained with different methods could be a good measure of robustness of phylogenetic reconstruction. The analysis presented here shows that even the most robust methods, such as Maximum Likelihood or Bayesian Inference, may show conflicting topologies, and selecting one over another may lead to wrong conclusions. Increasing the number of informative sites or, conversely, removing saturated fragments from the sequences leads to the inference of consensus topology independent of the methodology applied. Of the different tree-building algorithms, BI was less sensitive to the inclusion/ exclusion of individual genes from the analyses compared to the three other algorithms. Maximum Parsimony provided the least resolution of unweighted mitogenome data, and NJ and ML produced the phylogenies complementary to each other but not to MP and BI, and these algorithms are sensitive to the inclusion/ exclusion of individual mitochondrial genes.

Sequences of individual mitochondrial genes are not sufficiently informative to produce a phylogeny fully resolved and congruent to a whole mitogenome-based phylogeny. Each gene examined in our analyses exhibited from one to seven unresolved nodes, with ML analysis showing less resolution than BI. However, significant deviations from the mitogenome topology were more common when BI was applied. Resolution of nodes between closely related species was less dependent on the gene than were basal nodes. The 16S RNA gene showed only one deviation from the mitogenome topology (concerning the basal branching between *D. parvula* and other *Darevskia*, vs. its clustering with the *rudis* clade), and independent of the methodology applied. This gene also showed the lowest saturation index. The Cytochrome b gene was the second most informative gene, with no significant deviation from the whole mitogenome topology and one unresolved position when the ML method was used; with BI, it showed one significant deviation from the whole mitogenome phylogeny. The combination of these two genes produced a topology that is fully congruent with the whole mitogenome topology.

## 5. Conclusions

There are some indicators that force us to critically evaluate many phylogenetic studies, even ones that result from much analytical work and large data sets. Even complete mitochondrial

sequences may leave unresolved questions if substantial portions of these sequences are saturated. The second lesson is that even small parts of mitochondrial genome, if the sequenced genes are logically selected and adjusted to the relatedness level among the studied species, provide adequate phylogeny. Specifically, the 16S mitochondrial gene (1566 bp) appears to be sufficient for reconstructing species-level phylogenies within genera of true lizards. One can gain some useful information from sequencing portions of Cytochrome b gene, together with 16S. If high-throughput sequencing is not available, it may be useful to combine sequences of mitochondrial and nuclear genes, as Meiklejohn et al. [77] recommend; however, when closely related species are studied, the nuclear and mitochondrial phylogenies may not coincide due to incomplete lineage sorting or gene introgression, and mitochondrial phylogenies retain their importance for reconstructing matrilineal phylogeny.

## Supporting information

**S1 Fig. Substitution saturation plot for individual genes of mitochondrial genome.** Nucleotide transitions and transversions versus divergence. The vertical axes is for the observed proportion of transitions (s) and transversions (v), respectively. Gene name is indicated on the each graf, genetic distance was applied according Table 1.
(DOCX)

## Acknowledgments

The NGS of lizard samples were carried out at R. Lugar Genome Center, National Center of Disease Control and Public Health Tbilisi, Georgia. Bioinformatic analysis of the data was performed under supervision of Todd Castoe (University of Texas at Arlington, TX). Bernhard Misof, Simon Kaffer and Matthew Fujita provided valuable suggestions during data analysis. We are very grateful to Nato Kotaria for her help with lab work. We thank our co-workers Alexander Gavashelishvili, Levan Mumladze. Mariam Gabelaia, Natia Barateli, and Giorgi Iankoshvili for their assistance in sample collection and Viktor Spannenberg for the additional samples. Ulrich Joger and two anonymous referees made helpful comments on the first draft of the Manuscript.

## Author Contributions

**Conceptualization:** Marine Murtskhvaladze, David Tarkhnishvili.

**Data curation:** Marine Murtskhvaladze, Adam Kotorashvili.

**Formal analysis:** Marine Murtskhvaladze.

**Funding acquisition:** Marine Murtskhvaladze.

**Methodology:** Marine Murtskhvaladze.

**Project administration:** Marine Murtskhvaladze.

**Resources:** Adam Kotorashvili.

**Software:** Marine Murtskhvaladze.

**Supervision:** David Tarkhnishvili.

**Validation:** David Tarkhnishvili.

**Visualization:** David Tarkhnishvili.

**Writing – original draft:** Marine Murtskhvaladze, David Tarkhnishvili.

**Writing – review & editing:** Marine Murtskhvaladze, David Tarkhnishvili, Cort L. Anderson.

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
