## [Decision Letter · Decision Letter 0]

22 Aug 2019

PONE-D-19-16166

PHYLOGENY OF CAUCASIAN ROCK LIZARDS (DAREVSKIA) AND OTHER TRUE LIZARDS BASED ON MITOGENOME ANALYSIS: RAPID RADIATION AND ALGORITHM-DEPENDENT POSITION OF BASAL NODES

PLOS ONE

Dear Dr Murtskhvaladze,

Thank you for submitting your manuscript to PLOS ONE. After careful consideration, we feel that it has merit but does not fully meet PLOS ONE’s publication criteria as it currently stands. Therefore, we invite you to submit a revised version of the manuscript that addresses the points raised during the review process.

Reviewer 1 judged your paper 'major revision needed' while reviewer 2 wanted to reject it completely. I follow here reviewer 1 in my decision. I do not think like reviewer 1 that phylogenies based on mitochondrial data alone have no value. But you should clearly discuss their limits. I also think that your data are worth to be publishes. But they are not complete enough for a good phylogeny of all Lacertini. Add more taxa or restrict your conclusions to those groups of which you have enough samples.

Please look carefully at all the points raised by both reviewers and re-write your manuscript completely. Discussion and conclusions should then be self-critical and show which conclusions are supported by your data and which not. I am looking forward to receive a new manuscript. 

We would appreciate receiving your revised manuscript by Oct 06 2019 11:59PM. To enhance the reproducibility of your results, we recommend that if applicable you deposit your laboratory protocols in protocols.io, where a protocol can be assigned its own identifier (DOI) such that it can be cited independently in the future. For instructions see: http://journals.plos.org/plosone/s/submission-guidelines#loc-laboratory-protocols

We look forward to receiving your revised manuscript.

Kind regards,

Ulrich Joger

Academic Editor

PLOS ONE

Journal Requirements:

3. Thank you for including an Ethics Statement with your submission. At this time, for purposes of reporting, we ask that you please also add your Ethics Statement to your Materials and Methods section.

Reviewers' comments:

Reviewer's Responses to Questions

**Comments to the Author**

1. Is the manuscript technically sound, and do the data support the conclusions?

Reviewer #1: Partly

Reviewer #2: No

2. Has the statistical analysis been performed appropriately and rigorously? 

Reviewer #1: Yes

Reviewer #2: Yes

3. Have the authors made all data underlying the findings in their manuscript fully available?

Reviewer #1: Yes

Reviewer #2: Yes

4. Is the manuscript presented in an intelligible fashion and written in standard English?

Reviewer #1: Yes

Reviewer #2: Yes

5. Review Comments to the Author

Reviewer #1: The manuscript entitled PHYLOGENY OF CAUCASIAN ROCK LIZARDS (DAREVSKIA) AND OTHER TRUE LIZARDS BASED ON MITOGENOME ANALYSIS: RAPID RADIATION AND ALGORITHM DEPENDENT POSITION OF BASAL NODES is nice contribution presenting whole mitogenome based phylogeny of Darevskia and related genera of true lizards. Authors have sequenced and assembled 16 new mitochondrial genomes of Darevskia species, thus making possible to build phylogeny of Lacertini genera and resolve relationships inside Darevskia genus. The major finding in addition to a sequenced new mitochondrial genomes are detailed and well grounded workflow of analysis of mitochondrial sequences including approach to minimize influence of substitution saturation. Unfortunately, these real advantages are not properly covered in the discussion and conclusions sections and instead too much space is devoted to discussion of topics which deserves to be a subject of separate research and instead has limited support from this particular study results, therefore the manuscript needs a major revision.

Authors did a good job comparing different approaches to whole mitogenome-based phylogeny reconstruction. One of the major findings is that saturation is the main obstacle in reconstruction of robust phylogeny and this could be overcome by careful selection of genes and outgroup or running analyses without outgroup at all. Unfortunately, this have not been reflected in the abstract. In addition, convergence of topologies obtained with different methods could be a good measure of robustness of phylogenetic reconstruction – authors should say it directly. Finally, I don’t see clear evidence of rapid radiation, just mitochodrial genome comes close to the limit of unambiguously reconstructing of ancient splits.

Introduction could be strengthen by adding information about possible flaws of mtDNA in phylogenetic reconstruction, namely only maternal inheritance, capture or lost by lineages via introgression, not-neutrality, functional ties with nuclear genes, possible presence of pseudogenes etc., and how at least some of these difficulties can be targeted when using complete mitochondrial genomes. Please, check these papers

Podnar, M., Haring, E., Pinsker, W., Mayer., W. 2007. Unusual origin of a nuclear pseudogene in the Italian wall lizard: Intergenomic and interspecific transfer of a large section of the mitochondrial genome in the genus Podarcis (Lacertidae). J. Mol. Evol. 64, 308-320.

Godinho, R., Crespo, E.G., Ferrand, M., 2008. The limits of mtDNA phylogeography: complex patterns of population history in a highly structured Iberian lizard are only revealed by the use of nuclear markers. Molecular Ecology. 17, 4670–4683.

https://www.pnas.org/content/99/25/16122?cited-by=yesl99%2F25%2F16122&utm_source=TrendMD&utm_medium=cpc&utm_campaign=Proc_Natl_Acad_Sci_U_S_A_TrendMD_0

https://www.biorxiv.org/content/10.1101/588756v2.abstract

I think the manuscript should be substantially revised to be more concise, and keep track on the main ideas whose are supported by results. This includes changing title of the paper to more appropriate (rapid radiation is not the only possible reason for absence of strong support of basal nodes; perhaps loosing of phylogenetic signal and non-monotonous rates of molecular evolution among species could also contribute).

Other remarks to the text (unfortunately, lines numbering in PDF was missing):

“inter-

generic phylogenies.” - there is extra return in Abstract

“but BI and ML topologies support basal position of D. parvula from the western Lesser Caucasus.” - did you mean BI and MP?

“All topologies shifted the phylogenetic position” - did you mean altered, shuffled?

In Methods “The sum of the conflicting, coinciding, and unsupported nodes was considered a measure of similarity between the compared trees” - haven't you consider plotting some of these numbers against saturation indicator to show it’s influence?

The problem of reconstruction of phylogeny using saturated mitochondrial genes seems not new – see for example https://www.ncbi.nlm.nih.gov/pubmed/11975355
https://www.ncbi.nlm.nih.gov/pubmed/29669515 and it should be discussed

Fig. 2 legend “Posterior probabilities left of the nodes” does not seem grammatically correct

“However, genus-level topologies were still different (results not shown).” - you have Supplementary files where you can show these results.

There is subsection “Differences from earlier findings” in Discussion. I think it should be shortened – obviously lots of reconstructions had been made, there is no need to discuss differences in such detailed way.

I am surprised not to see molecular dating reconstruction. Authors definitely have very good dataset for that, moreover, they are studying one taxonomic group, which is very much the group of their expertise – why not? It would add value to the publication and justify discussion of biogeographic scenario if it will persist in the final version.

“In the Western Lesser Caucasus area, D. parvula coexists with the representatives of the

next basal clade of Darevskia: the clade ‘D. rudis’ (in sense of (1). Consequently, this area is the

most likely ancestral for all Darevskia. In the original time of divergence, this was the north-east-

ern part of the Anatolian landmass, and the separation could have happened between the water-

sheds separating the upper reaches of the Kura (Mtkvari), Chorokhi (Coruh), and Euphrates Rivers,

due to the growth of xeric and less suitable areas between the lizard habitats.” - Authors may use more formal approach to the ancestral range calculation. Currently, this discussion looks speculative, especially the part with discussion of climate of watersheds. 20 Mya the area could look different due to massive transgressions-regression cycles there – see Popov, Rögl, Rozanov_2004_Lithological-Paleogeographic Maps of Paratethys Cour. Forsch.-Inst. Senckenberg, 2004. — 250 p. Authors definitely must to do additional analyses inferring split time, ancestral areas, ecological features of species of Darevskia genus (ground dwelling versus rock dwelling) and rewrite correspondent sections of manuscript or omit these topics. Actually I would suggest to concentrate here only on molecular dating, since the last two issues would benefit from additional taxa, which are not included in this study.

Conclusions must be rewritten according to the new version of manuscript. As an example only, currently “The first is that constructing phylogenies based on the analysis of high volumes of various genetic data is not the best approach for reconstructing species-level phylogenies.” is highly ambiguous; “Phylogenetic studies devoted to a specific taxonomic group, such as genera or subfamily, are more trustful for resolving species-level taxonomies than meta-analyses comprising large number of taxa with different evolutionary distances from each other.” is trivial and too general and so on. The last part of last sentence “however some elements of topology are unlikely to be finally resolved even if the complete mitochondrial genome is studied, and, if high-throughput sequencing is not available, it is more useful to combine sequences of mitochondrial and nuclear genes, as (66) recommend.” is merely speculative and is not conclusion of this research.

Reviewer #2: In the present manuscript the authors sequence 16 mitochondrial genomes of the lacertid genus Darevskia and use them to reach conclusions of the phylogenetic relationships within this genus and, together with GenBank sequences, to infer the phylogenetic relationships within Lacertini. Although sequencing 16 mitochondrial genomes is a good effort it is certainly not the way forward to solve evolutionary relationships in the XXI century. The mtDNA is very long an can provide quite a lot of information but the main drawback is that it represents a single locus. In fact, mtDNA is a very particular locus with exclusively maternal inheritance and with a high evolutionary rate. Aspects that have proven very good for resolving many evolutionary relationships but that have also been shown to recover wrong phylogenetic relationships in many occasions. For this reason, in the last 10 years it has been very usual to apply a multilocus approach to phylogenetic inference, first with a few loci (2-3) but latter on with more (5-8) and more recently with hundreds or even thousands of loci. Of course, missing data is still an issue but multilocus phylogenies are certainly the way forward to obtaining the correct evolutionary relationships of taxa. Especially on these groups in which the phylogenetic relstionships are contentious. NGS techniques have opened new doors to dig further into the phylogenetic relationships of difficult groups (including Lacertini) using many independent loci. To suddenly apply these NGS techniques to obtain long stretches of a highly variable and maternally inherited single locus is like going back to the beginning of phylogenetic inference. In fact, one of the conclusions of the present manuscript is that just to genes, 16S and cytb, give the same (or very similar) results than the whole mitogenomes. Phylogenies of 16S and cytb have been around for more than 20 years and luckily technical and analytical advances occurred in the last 10 years have allowed a leap forward towards multilocus phylogenies, a much better approach than single locus phylogenies like the one presented in the present study.

In the manuscript the authors use a largely incomplete phylogeny of Lacertidae to discuss the phylogenetic relationships within Lacertini. The results presented are completely uninteresting and definitively not worth publishing. The authors should delete this whole section from the paper. There have been recent studies using multilocus data that have not been included in the manuscript (Mendes et al. 2016 Molecular Phylogenetics and Evolution 100 (2016) 254–267) and this makes the whole discussion comparing to papers by Fu et al. and Harris et al from 1997 and 1998 and other older papers irrelevant and uninteresting. The authors are missing many Lacertini genera and therefore they should not use their incomplete dataset to discuss the phylogenetic relationships within this very relevant group. Doing that is a superfluous and confusing exercise that is, again, not worth publishing. After Mendes et al. 2016 the phylogeny of Lacertini is more or less fixed until new studies using thousands of loci obtained using NGS will be published in 2020. Datasets should be developed to answer specific questions and this partial mitogenomic dataset is definitively not suitable to address the phylogeny of Lacertini.

Across the whole manuscript the authors use the word “basal” incorrectly. I urge the authors to read this paper very carefully and correct the way they discuss phylogenies.

https://www.csun.edu/~dgray/Evol322/OmlandTreeThinking.pdf

The exercise within Darevskia is interesting but again it is restricted to half of the described species. In fact, they only include species within Georgia.

Minor comments:

P12: Neither in the Fig. legend or in the figure itself the authors indicate Georgia. They should indicate where is Georgia.

P14: The D. parvula is most probably D. adjarica (previously D. parvula adjarica). See https://biotaxa.org/Zootaxa/article/view/zootaxa.4472.1.3

P14: D. clarcorum is wrongly written in several places in the text (OK in the map). It is d. clarkorum, with “k”.

Pag 14: “E. multcocellata” is E. multiocellata

Pag 17: in several places they use “polytiomies”, correct to “polytomies”

Pag 21: Legend for Fig. 4 “genes” is repeated twice

Within Darevskia, “parvula” (or better adjarica; see Arribas et al. 2018) is sister taxon to the remaining Darevskia included in the study. Despite they present it as a new finding, it had been already published by Fu et al. (1997) and mentioned by Arribas et al (2018).

Please, check this paragraph from Arribas et al. (2018):

“Grechko et al. (1997) found that D. parvula was sister taxon to all the other Darevskia when studied using “DNA Taxon prints” (Restriction endonuclease analysis of Highly Repetitive DNA). Ciobanu et al. (2002, 2003) and Grechko et al. (2006) also studied Satellite DNA in Darevskia and found a relatively isolated position of D. parvula in which only one type of Satellite DNA predominate (CLsatIII), and as inhabitants of one of the glacial refuges in the extremes of the Caucasus (as D. clarkorum/dryada) in the west and D. chlorogaster in the southeast, they argue that possibly they resemble the ancestors of the group. The presence of these same microsatellites in the “mixta-group” (clarkorum, caucasica, daghestanica, dryada) casts some doubts on the basal association of D. parvula with the rudis group as stated by means of mtDNA (see above). For Grechko et al. (2006), D. parvula would be outside these two clades, but hybridized in an early instance of its diversification with the ancestors of both clades.”

-At the end of page 17 they comment: “We did not include D. alpina in our analysis, however, this species is matrilineally very close to D. saxicola, and may be of hybrid origin”. What is possibly of hybrid origin is not the species but the individual sewuenced by Murphy et al. To extrapolate it to the whole species is a bit excessive.

Page 27 etc. As mentioned in the general comments above, the manuscript is not suitable for addressing the phylogenetic relationships and/or the biogeography of lacertini. It is missing many taxa, it uses a single locus and therefore it is wrong and confusing. I would suggest the authors to refrain from publishing these results on the phylogeny and biogeography of Lacertini

6. PLOS authors have the option to publish the peer review history of their article (what does this mean?). If published, this will include your full peer review and any attached files.

Reviewer #1: No

Reviewer #2: No

---

## [Author Response · Author response to Decision Letter 0]

7 Oct 2019

Reviewer 1 judged your paper 'major revision needed' while reviewer 2 wanted to reject it completely. I follow here reviewer 1 in my decision. I do not think like reviewer 1 that phylogenies based on mitochondrial data alone have no value. But you should clearly discuss their limits. I also think that your data are worth to be publishes. But they are not complete enough for a good phylogeny of all Lacertini. Add more taxa or restrict your conclusions to those groups of which you have enough samples.

- In the revised version of the manuscript, we restrict ourselves to the phylogeny of Darevskia and included only few critical points about the outgroups, not pretending on the reconstruction of phylogeny of true lizards.

Please look carefully at all the points raised by both reviewers and re-write your manuscript completely. Discussion and conclusions should then be self-critical and show which conclusions are supported by your data and which not. I am looking forward to receive a new manuscript. 

- The MS is rewritten almost completely following the reviewers comments, and some comments are discussed below.

5. Review Comments to the Author

Reviewer #1: The manuscript entitled PHYLOGENY OF CAUCASIAN ROCK LIZARDS (DAREVSKIA) AND OTHER TRUE LIZARDS BASED ON MITOGENOME ANALYSIS: RAPID RADIATION AND ALGORITHM DEPENDENT POSITION OF BASAL NODES is nice contribution presenting whole mitogenome based phylogeny of Darevskia and related genera of true lizards. Authors have sequenced and assembled 16 new mitochondrial genomes of Darevskia species, thus making possible to build phylogeny of Lacertini genera and resolve relationships inside Darevskia genus. The major finding in addition to a sequenced new mitochondrial genomes are detailed and well-grounded workflow of analysis of mitochondrial sequences including approach to minimize influence of substitution saturation. Unfortunately, these real advantages are not properly covered in the discussion and conclusions sections and instead too much space is devoted to discussion of topics which deserves to be a subject of separate research and instead has limited support from this particular study results, therefore the manuscript needs a major revision.

-the authors tried to shift the emphasis in the discussion substantially, in particular they reduce the parts not directly linked with the findings.

Authors did a good job comparing different approaches to whole mitogenome-based phylogeny reconstruction. One of the major findings is that saturation is the main obstacle in reconstruction of robust phylogeny and this could be overcome by careful selection of genes and outgroup or running analyses without outgroup at all. Unfortunately, this have not been reflected in the abstract.

-we tried to discuss this in more detail in the revision, and shift the focus away from the discussion about geological past etc. (only retained in part that is directly related to the genus Darevskia).

In addition, convergence of topologies obtained with different methods could be a good measure of robustness of phylogenetic reconstruction – authors should say it directly. 

-done

Finally, I don’t see clear evidence of rapid radiation, just mitochodrial genome comes close to the limit of unambiguously reconstructing of ancient splits.

-agree; the discussion on the potential rapid radiation removed as unnecessary. 

Introduction could be strengthen by adding information about possible flaws of mtDNA in phylogenetic reconstruction, namely only maternal inheritance, capture or lost by lineages via introgression, not-neutrality, functional ties with nuclear genes, possible presence of pseudogenes etc., and how at least some of these difficulties can be targeted when using complete mitochondrial genomes. Please, check these papers

Podnar, M., Haring, E., Pinsker, W., Mayer., W. 2007. Unusual origin of a nuclear pseudogene in the Italian wall lizard: Intergenomic and interspecific transfer of a large section of the mitochondrial genome in the genus Podarcis (Lacertidae). J. Mol. Evol. 64, 308-320.

Godinho, R., Crespo, E.G., Ferrand, M., 2008. The limits of mtDNA phylogeography: complex patterns of population history in a highly structured Iberian lizard are only revealed by the use of nuclear markers. Molecular Ecology. 17, 4670–4683.

https://www.pnas.org/content/99/25/16122?cited-by=yesl99%2F25%2F16122&utm_source=TrendMD&utm_medium=cpc&utm_campaign=Proc_Natl_Acad_Sci_U_S_A_TrendMD_0

https://www.biorxiv.org/content/10.1101/588756v2.abstract

-these issues are addressed in the revised introduction

I think the manuscript should be substantially revised to be more concise, and keep track on the main ideas whose are supported by results. This includes changing title of the paper to more appropriate (rapid radiation is not the only possible reason for absence of strong support of basal nodes; perhaps loosing of phylogenetic signal and non-monotonous rates of molecular evolution among species could also contribute).

- agreed, the title is changed, and the relevant issues addressed in the revision

Other remarks to the text (unfortunately, lines numbering in PDF was missing):

-lines numbering inserted in the revision

“inter-generic phylogenies.” - there is extra return in Abstract

“but BI and ML topologies support basal position of D. parvula from the western Lesser Caucasus.” - did you mean BI and MP?

-of course, thank you for the correction. 

“All topologies shifted the phylogenetic position” - did you mean altered, shuffled?

- altered; corrected

In Methods “The sum of the conflicting, coinciding, and unsupported nodes was considered a measure of similarity between the compared trees” - haven't you consider plotting some of these numbers against saturation indicator to show it’s influence?

-thank you for this suggestion! In the revision, the correlation between the conflicting node number and saturation index is discussed.

The problem of reconstruction of phylogeny using saturated mitochondrial genes seems not new – see for example https://www.ncbi.nlm.nih.gov/pubmed/11975355
https://www.ncbi.nlm.nih.gov/pubmed/29669515 and it should be discussed

- we addressed the issue in the revision

Fig. 2 legend “Posterior probabilities left of the nodes” does not seem grammatically correct

-corrected

“However, genus-level topologies were still different (results not shown).” - you have Supplementary files where you can show these results.

-we don’t discuss genus-level topologies in the revision, with the exception of two important patterns: basal Takydromus and sister status of Zootoka and Podarcis.

There is subsection “Differences from earlier findings” in Discussion. I think it should be shortened – obviously lots of reconstructions had been made, there is no need to discuss differences in such detailed way.

- the discussion is reduced substantially, according to this recommendation. Besides, we minimized the statements on the Lacertini phylogeny, considering the recommendation of the 2nd referee and the editor, although retained few points that we think are important and well-supported. 

I am surprised not to see molecular dating reconstruction. Authors definitely have very good dataset for that, moreover, they are studying one taxonomic group, which is very much the group of their expertise – why not? It would add value to the publication and justify discussion of biogeographic scenario if it will persist in the final version.

-we prefer to be careful about the dating, since there is very broad range of calibrating molecular clock for lacertids by different authors. However, following the suggestion of the referee, we used two alternative datings, more and less conservative, referring on two important geologicasl events shaping the region: middle Miocene climate transition and Messinian salinity crisis. The respective datings are discussed and illustrated. 

“In the Western Lesser Caucasus area, D. parvula coexists with the representatives of the

next basal clade of Darevskia: the clade ‘D. rudis’ (in sense of (1). Consequently, this area is the

most likely ancestral for all Darevskia. In the original time of divergence, this was the north-east-

ern part of the Anatolian landmass, and the separation could have happened between the water-

sheds separating the upper reaches of the Kura (Mtkvari), Chorokhi (Coruh), and Euphrates Rivers,

due to the growth of xeric and less suitable areas between the lizard habitats.” - Authors may use more formal approach to the ancestral range calculation. Currently, this discussion looks speculative, especially the part with discussion of climate of watersheds. 20 Mya the area could look different due to massive transgressions-regression cycles there – see Popov, Rögl, Rozanov_2004_Lithological-Paleogeographic Maps of Paratethys Cour. Forsch.-Inst. Senckenberg, 2004. — 250 p. Authors definitely must to do additional analyses inferring split time, ancestral areas, ecological features of species of Darevskia genus (ground dwelling versus rock dwelling) and rewrite correspondent sections of manuscript or omit these topics. Actually I would suggest to concentrate here only on molecular dating, since the last two issues would benefit from additional taxa, which are not included in this study.

-we followed this suggestion working on the revision.

Conclusions must be rewritten according to the new version of manuscript. As an example only, currently “The first is that constructing phylogenies based on the analysis of high volumes of various genetic data is not the best approach for reconstructing species-level phylogenies.” is highly ambiguous;

-Conclusions section re-written

 “Phylogenetic studies devoted to a specific taxonomic group, such as genera or subfamily, are more trustful for resolving species-level taxonomies than meta-analyses comprising large number of taxa with different evolutionary distances from each other.” is trivial and too general and so on. 

-agree, changed

The last part of last sentence “however some elements of topology are unlikely to be finally resolved even if the complete mitochondrial genome is studied, and, if high-throughput sequencing is not available, it is more useful to combine sequences of mitochondrial and nuclear genes, as (66) recommend.” is merely speculative and is not conclusion of this research.

-removed

Reviewer #2: In the present manuscript the authors sequence 16 mitochondrial genomes of the lacertid genus Darevskia and use them to reach conclusions of the phylogenetic relationships within this genus and, together with GenBank sequences, to infer the phylogenetic relationships within Lacertini. Although sequencing 16 mitochondrial genomes is a good effort it is certainly not the way forward to solve evolutionary relationships in the XXI century. The mtDNA is very long an can provide quite a lot of information but the main drawback is that it represents a single locus. In fact, mtDNA is a very particular locus with exclusively maternal inheritance and with a high evolutionary rate. Aspects that have proven very good for resolving many evolutionary relationships but that have also been shown to recover wrong phylogenetic relationships in many occasions. For this reason, in the last 10 years it has been very usual to apply a multilocus approach to phylogenetic inference, first with a few loci (2-3) but latter on with more (5-8) and more recently with hundreds or even thousands of loci. Of course, missing data is still an issue but multilocus phylogenies are certainly the way forward to obtaining the correct evolutionary relationships of taxa. Especially on these groups in which the phylogenetic relstionships are contentious. NGS techniques have opened new doors to dig further into the phylogenetic relationships of difficult groups (including Lacertini) using many independent loci. To suddenly apply these NGS techniques to obtain long stretches of a highly variable and maternally inherited single locus is like going back to the beginning of phylogenetic inference. In fact, one of the conclusions of the present manuscript is that just to genes, 16S and cytb, give the same (or very similar) results than the whole mitogenomes. Phylogenies of 16S and cytb have been around for more than 20 years and luckily technical and analytical advances occurred in the last 10 years have allowed a leap forward towards multilocus phylogenies, a much better approach than single locus phylogenies like the one presented in the present study.

- We know that it is possible to sequence nuclear genes and also use genomic data. However, (1) multilocus approach has its own shortages for reconstruction of low-level phylogenies, where gene introgression commonly takes place, or incomplete lineage sorting may be the case. In fact, mitochondrial DNA phylogeny may not completely coincide with species-level phylogeny but its interpretation is straightforward – this is phylogeny of maternal lineages. This is indicated in the text. (2) as mentioned in the paper, there are hundreds of DNA laboratories worldwide that are and will be during quite long time rely on PCR based methods and individual gene sequences. If answers on some specific questions can be found with less equipment and cheaper techniques, we should not reject this opportunity. 

In the manuscript the authors use a largely incomplete phylogeny of Lacertidae to discuss the phylogenetic relationships within Lacertini. The results presented are completely uninteresting and definitively not worth publishing. The authors should delete this whole section from the paper. There have been recent studies using multilocus data that have not been included in the manuscript (Mendes et al. 2016 Molecular Phylogenetics and Evolution 100 (2016) 254–267) and this makes the whole discussion comparing to papers by Fu et al. and Harris et al from 1997 and 1998 and other older papers irrelevant and uninteresting. The authors are missing many Lacertini genera and therefore they should not use their incomplete dataset to discuss the phylogenetic relationships within this very relevant group. Doing that is a superfluous and confusing exercise that is, again, not worth publishing. After Mendes et al. 2016 the phylogeny of Lacertini is more or less fixed until new studies using thousands of loci obtained using NGS will be published in 2020. Datasets should be developed to answer specific questions and this partial mitogenomic dataset is definitively not suitable to address the phylogeny of Lacertini.

-the paper of Mendes et al. is cited in the revision. The discussion on the inter-generic relationships of Lacertini is strongly reduced. 

Across the whole manuscript the authors use the word “basal” incorrectly. I urge the authors to read this paper very carefully and correct the way they discuss phylogenies.

https://www.csun.edu/~dgray/Evol322/OmlandTreeThinking.pdf

-The authors are familiar with the basics of cladistics philosophy, and they did not use the word “basal” in sense of “ancestral” which seems to be the concern of this reviewer--Basal is used in sense of the branch or clade in the phylogeny most distant (and earliest separated) from the rest of the evolutionary lineages. We do not agree with the reviewer that this term is used incorrectly throughout the manuscript. 

The exercise within Darevskia is interesting but again it is restricted to half of the described species. In fact, they only include species within Georgia.

- The species set included in the analysis represents all clades and major groups of Darevskia, except group D. defillippi. There are species of the genus found far south, west, and east from the sampling area, however, they are either found in Georgia as well, or their closest relatives are found there. The only group of species not found in Georgia is D. chlorogaster, and the sample of this species collected from SE Azerbaijan is included. Because all of the major clades are well represented, and because relationships of the more terminal taxa are not at issue, we feel that the species presented in this analysis provide an optimal assortment of Darevskia taxa, commensurate with our limited resources and time.

Minor comments:

P12: Neither in the Fig. legend or in the figure itself the authors indicate Georgia. They should indicate where is Georgia.

-corrected

P14: The D. parvula is most probably D. adjarica (previously D. parvula adjarica). See https://biotaxa.org/Zootaxa/article/view/zootaxa.4472.1.3

-according to the Reptile Database (http://www.reptile-database.org/) it is still D. p. adjarica, we rely on this nomenclature.

P14: D. clarcorum is wrongly written in several places in the text (OK in the map). It is d. clarkorum, with “k”.

-corrected

Pag 14: “E. multcocellata” is E. multiocellata

-corrected

Pag 17: in several places they use “polytiomies”, correct to “polytomies”

-corrected

Pag 21: Legend for Fig. 4 “genes” is repeated twice

Within Darevskia, “parvula” (or better adjarica; see Arribas et al. 2018) is sister taxon to the remaining Darevskia included in the study. Despite they present it as a new finding, it had been already published by Fu et al. (1997) and mentioned by Arribas et al (2018).

Please, check this paragraph from Arribas et al. (2018):

“Grechko et al. (1997) found that D. parvula was sister taxon to all the other Darevskia when studied using “DNA Taxon prints” (Restriction endonuclease analysis of Highly Repetitive DNA). Ciobanu et al. (2002, 2003) and Grechko et al. (2006) also studied Satellite DNA in Darevskia and found a relatively isolated position of D. parvula in which only one type of Satellite DNA predominate (CLsatIII), and as inhabitants of one of the glacial refuges in the extremes of the Caucasus (as D. clarkorum/dryada) in the west and D. chlorogaster in the southeast, they argue that possibly they resemble the ancestors of the group. The presence of these same microsatellites in the “mixta-group” (clarkorum, caucasica, daghestanica, dryada) casts some doubts on the basal association of D. parvula with the rudis group as stated by means of mtDNA (see above). For Grechko et al. (2006), D. parvula would be outside these two clades, but hybridized in an early instance of its diversification with the ancestors of both clades.”

- Thank you for this comment. Fu et al. (1997) suggested basal position for D. praticola, and not D. parvula. Grechko et al. indeed hypothesized this based on the microsatellite and restriction endonuclease analysis, and we mention this in the revision. 

-At the end of page 17 they comment: “We did not include D. alpina in our analysis, however, this species is matrilineally very close to D. saxicola, and may be of hybrid origin”. What is possibly of hybrid origin is not the species but the individual sewuenced by Murphy et al. To extrapolate it to the whole species is a bit excessive.

- the statement is re-worded. 

Page 27 etc. As mentioned in the general comments above, the manuscript is not suitable for addressing the phylogenetic relationships and/or the biogeography of lacertini. It is missing many taxa, it uses a single locus and therefore it is wrong and confusing. I would suggest the authors to refrain from publishing these results on the phylogeny and biogeography of Lacertini

-as said above, the respective sections are removed or greatly reduced

---

## [Decision Letter · Decision Letter 1]

31 Dec 2019

PONE-D-19-16166R1

PHYLOGENY OF CAUCASIAN ROCK LIZARDS (DAREVSKIA) AND SOME OTHER TRUE LIZARDS BASED ON MITOGENOME ANALYSIS: OPTIMISATION OF THE ALGORITHMS AND GENE SELECTION

PLOS ONE

Dear Dr Murtskhvaladze,

Thank you for submitting your manuscript to PLOS ONE. After careful consideration, we feel that it has merit but does not fully meet PLOS ONE’s publication criteria as it currently stands. Therefore, we invite you to submit a revised version of the manuscript that addresses the points raised during the review process.

Unfortunately, you did not obey all the critical points raised by the reviewers. Please consider their critique more seriously. Especially you should consider all recent literature on lacertid phylogeny and on Darevskia.

One reviewer even speculated that you sent an earlier version of the manuscript again. You should really re-write it according to the suggestions below.

We would appreciate receiving your revised manuscript by Feb 14 2020 11:59PM. To enhance the reproducibility of your results, we recommend that if applicable you deposit your laboratory protocols in protocols.io, where a protocol can be assigned its own identifier (DOI) such that it can be cited independently in the future. For instructions see: http://journals.plos.org/plosone/s/submission-guidelines#loc-laboratory-protocols

We look forward to receiving your revised manuscript.

Kind regards,

Ulrich Joger

Academic Editor

PLOS ONE

Reviewers' comments:

Reviewer's Responses to Questions

**Comments to the Author**

1. If the authors have adequately addressed your comments raised in a previous round of review and you feel that this manuscript is now acceptable for publication, you may indicate that here to bypass the “Comments to the Author” section, enter your conflict of interest statement in the “Confidential to Editor” section, and submit your "Accept" recommendation.

Reviewer #1: (No Response)

Reviewer #2: (No Response)

2. Is the manuscript technically sound, and do the data support the conclusions?

Reviewer #1: Yes

Reviewer #2: Partly

3. Has the statistical analysis been performed appropriately and rigorously? 

Reviewer #1: Yes

Reviewer #2: Yes

4. Have the authors made all data underlying the findings in their manuscript fully available?

Reviewer #1: Yes

Reviewer #2: Yes

5. Is the manuscript presented in an intelligible fashion and written in standard English?

Reviewer #1: Yes

Reviewer #2: Yes

6. Review Comments to the Author

Reviewer #1: Authors formally agreed with criticism but did not change the text appropriately in several instances and, as for me, major revision had not been done. The manuscript did not impove much, authors must rewrite it completely. Instances below makes me believe that it might be wrong version of the manuscript uploaded: even E. multcocellata was not corrected.

And some indicators of the absence of important changes.

Rapid radiation is still listed already in Abstract - “It appears that rapid radiation of lacertid genera complicates reconstruction of reliable phylogeny, even when using complete mitochondrial genome sequence”.). Line numbering is missing again.

“but BI and ML topologies support basal position of D. parvula from the western Lesser

Caucasus.” - did you mean BI and MP?

-of course, thank you for the correction.

“All topologies shifted the phylogenetic position” - did you mean altered, shuffled?”

– Authors agreed but did not change – was I wrong?

“In Methods “The sum of the conflicting, coinciding, and unsupported nodes was

considered a measure of similarity between the compared trees” - haven't you consider

plotting some of these numbers against saturation indicator to show it’s influence?

-thank you for this suggestion! In the revision, the correlation between the conflicting

node number and saturation index is discussed.”

- Same, can’t see any changes in the manuscript. Numbers in the Table 1 are hard to get and interprete.

“The problem of reconstruction of phylogeny using saturated mitochondrial genes

seems not new – see for example https://www.ncbi.nlm.nih.gov/pubmed/11975355

https://www.ncbi.nlm.nih.gov/pubmed/29669515 and it should be discussed

- we addressed the issue in the revision”

Where and how?

Reviewer #2: As I mentioned in my previous review, I consider the present work more confusing than helpful. It is a pity that the authors have invested so much time, effort and money to produce mitogenomes instead of producing nuclear data. In any case, at least in the revised version the authors have shortened the discussion on the phylogenetic relationships of Lacertini and restrict their discussion to Darevskia.

The authors do not use line numbers, which greatly difficult the review process.

I already commented in many aspects of the manuscript in my previous revision. Some f the suggestions were taken into account and some not. I do not agree with the answers of the reviewers to some of the comments. I am not going to insist/argue because it is not my paper.

Major comments:

The bibliography is not up-to-date with Lacetid phylogenetics including Darevskia phylogenetics. I already had to point out to some key studies in the previous revision.

In this occasion, the authors do not include in their discussion the recent paper by Garcia-Porta et al. 2019.

https://www.nature.com/articles/s41467-019-11943-x

This is a very relevant paper and the authors should have a careful look to the Lacertid phylogenies that are presented.

See the main phylogenies from the paper and also the calibrted phylogenies presented in Supplementary figures 10 and 11 including many representatives of the genus Dareskia.

The results of this paper should be discussed in the present manuscript. The calibrations are very well done.

Minor changes:

There is an “and” in italics

“FInder2” Finder2

“within the topology of genus….” within the topology of the genus

Why do they use “Zootoka”? it is Zootoca

“placement” placement?

“all analysis” all analyses.

7. PLOS authors have the option to publish the peer review history of their article (what does this mean?). If published, this will include your full peer review and any attached files.

Reviewer #1: No

Reviewer #2: No

---

## [Author Response · Author response to Decision Letter 1]

6 Jan 2020

Dear Editor,

First of all, the authors must greatly excuse for the inconvenience that happened during the last resubmission. The first author was confused and, unfortunately, along with the corrected version marked with the “track changes” option, uploaded partly corrected version of the manuscript. This caused unfortunate consequences, including the time of the referees spent for re-checking the incompletely revised version. If you could accept our excuses please consider the reviewed, re-written version of the manuscript that was largely completed by the beginning of October, 2019. Now, however, a number of additional changes, both in the text and Fig. 6 were included, first of all those that became necessary after publication of the paper of Garcia-Porta et al. This paper leaded us to some further changes of the already reviewed version of the manuscript. Some other slight modifications are included as well. Replies to the latest comments are included immediately after this letter, and below I retained the replies to earlier comments to the first version of the manuscript.

With kind regards and excuses, David Tarkhnishvili

….Unfortunately, you did not obey all the critical points raised by the reviewers. Please consider their critique more seriously. Especially you should consider all recent literature on lacertid phylogeny and on Darevskia.One reviewer even speculated that you sent an earlier version of the manuscript again. You should really re-write it according to the suggestions below.

- please consider the letter/ explanation / excuses above

6. Review Comments to the Author

Reviewer #1: Authors formally agreed with criticism but did not change the text appropriately in several instances and, as for me, major revision had not been done. The manuscript did not impove much, authors must rewrite it completely. Instances below makes me believe that it might be wrong version of the manuscript uploaded: even E. multcocellata was not corrected.

- please consider the author’s excuses for uploading incompletely revised manuscript in October, and see the corrected version.

Rapid radiation is still listed already in Abstract - “It appears that rapid radiation of lacertid genera complicates reconstruction of reliable phylogeny, even when using complete mitochondrial genome sequence”.). 

- Not any more, not in the completely revised version.

Line numbering is missing again.

- Not any more.

“but BI and ML topologies support basal position of D. parvula from the western Lesser

Caucasus.” - did you mean BI and MP?

-of course, thank you for the correction.

“All topologies shifted the phylogenetic position” - did you mean altered, shuffled?”

– Authors agreed but did not change – was I wrong?

- Changes included in the current version.

“In Methods “The sum of the conflicting, coinciding, and unsupported nodes was

considered a measure of similarity between the compared trees” - haven't you consider

plotting some of these numbers against saturation indicator to show it’s influence?

-thank you for this suggestion! In the revision, the correlation between the conflicting

node number and saturation index is discussed.”

- Same, can’t see any changes in the manuscript. Numbers in the Table 1 are hard to get and interprete.

- there is not correlation across all studied genes; however 16 S that has the highest valie of the saturation indicator also has the lowest number of conflicting/ unresolved nodes; and ATP8, with the lowest saturation indicator, has the most discrepancies from the mitogenome-based topology, this is explained in the current revision. We retained Table 1 but further simplified it. 

“The problem of reconstruction of phylogeny using saturated mitochondrial genes

seems not new – see for example https://www.ncbi.nlm.nih.gov/pubmed/11975355

https://www.ncbi.nlm.nih.gov/pubmed/29669515 and it should be discussed

- we addressed the issue in the revision”

Where and how?

- mentioned in the revised introduction.

Reviewer #2: As I mentioned in my previous review, I consider the present work more confusing than helpful. It is a pity that the authors have invested so much time, effort and money to produce mitogenomes instead of producing nuclear data. In any case, at least in the revised version the authors have shortened the discussion on the phylogenetic relationships of Lacertini and restrict their discussion to Darevskia.

The authors do not use line numbers, which greatly difficult the review process.

I already commented in many aspects of the manuscript in my previous revision. Some f the suggestions were taken into account and some not. I do not agree with the answers of the reviewers to some of the comments. I am not going to insist/argue because it is not my paper.

Major comments:

The bibliography is not up-to-date with Lacetid phylogenetics including Darevskia phylogenetics. I already had to point out to some key studies in the previous revision.

In this occasion, the authors do not include in their discussion the recent paper by Garcia-Porta et al. 2019.

https://www.nature.com/articles/s41467-019-11943-x

This is a very relevant paper and the authors should have a careful look to the Lacertid phylogenies that are presented.

See the main phylogenies from the paper and also the calibrted phylogenies presented in Supplementary figures 10 and 11 including many representatives of the genus Dareskia.

The results of this paper should be discussed in the present manuscript. The calibrations are very well done.

- thank you for this suggestion. We analyzed the paper of Garcia-Porta et al. thoroughly and inserted relevant changes in the ms. 

Minor changes:

There is an “and” in italics - corrected

“FInder2” Finder2 - corrected

“within the topology of genus….” within the topology of the genus

-the sentence absents from the revision

Why do they use “Zootoka”? it is Zootoca – corrected

“placement” placement?

“all analysis” all analyses.

- corrected

Replies to specific comments that are considering changes not inserted in the version uploaded in October.

The problem of reconstruction of phylogeny using saturated mitochondrial genes

seems not new – see for example https://www.ncbi.nlm.nih.gov/pubmed/11975355

https://www.ncbi.nlm.nih.gov/pubmed/29669515 and it should be discussed

- we addressed the issue in the revision”

Where and how?

- Introduction of the revision, lines 59-61

Reviewer #2: As I mentioned in my previous review, I consider the present work more confusing than helpful. It is a pity that the authors have invested so much time, effort and money to produce mitogenomes instead of producing nuclear data. 

- Again, cannot agree with this statement: In a new paper of Garcia-Porta et al. suggested by the reviewer (we appreciate much for providing link on this recent very important paper)the phylogenies based on the nuclear and mitochondrial genes fully coincide as stated by the authors. However, there are some differences from the phylogeny based on the full mitochondrial genome discussed in this paper, and we suppose that mitochondrial phylogeny in this paper is more representatives in terms of genes, although of course less representative considering the taxa included (some Darevskia species from southern Caspian among those).

Major comments:

The bibliography is not up-to-date with Lacetid phylogenetics including Darevskia phylogenetics. I already had to point out to some key studies in the previous revision.

In this occasion, the authors do not include in their discussion the recent paper by Garcia-Porta et al. 2019.

https://www.nature.com/articles/s41467-019-11943-x

This is a very relevant paper and the authors should have a careful look to the Lacertid phylogenies that are presented.

See the main phylogenies from the paper and also the calibrted phylogenies presented in Supplementary figures 10 and 11 including many representatives of the genus Dareskia.

The results of this paper should be discussed in the present manuscript. The calibrations are very well done.

- We appreciate a lot for this suggestion. Indeed, relevant changes/ specifications were included in the present version of the manuscript, both in the introduction and discussion parts.

Minor changes:

There is an “and” in italics

“FInder2” Finder2

-corrected

“within the topology of genus….” within the topology of the genus

- not in the revision

Why do they use “Zootoka”? it is Zootoca

- Corrected

“placement” placement?

-not in the revision

“all analysis” all analyses.

-corrected 

The letter sent with the first revision

Dear Editor,

Thank you for evaluating our manuscript on the phylogeny of Darevskia. We learned in details the comments of both reviewers and re-wrote the manuscript respectively; in particular, in the presentation of the results and the discussion we concentrated on the genus Darevskia rather than on the all Lacertini, although maintained a broad selection of true lizards as the outgroups; only short statements were maintained on the outgroups, comparing with the findings of the previous authors, specifically basal position of Takydromus within Lacertini and sister status for Podarcis and Zootoka. We also removed the discussion on the potential factors of divergence among the Lacertini genera. The title of the paper has been modified. 

Dear Dr Murtskhvaladze,

Thank you for submitting your manuscript to PLOS ONE. After careful consideration, we feel that it has merit but does not fully meet PLOS ONE’s publication criteria as it currently stands. Therefore, we invite you to submit a revised version of the manuscript that addresses the points raised during the review process.

- thank you. We substantially re-analysed and re-writed the manuscript according to the editor’s and the reviewer’s comments.

Reviewer 1 judged your paper 'major revision needed' while reviewer 2 wanted to reject it completely. I follow here reviewer 1 in my decision. I do not think like reviewer 1 that phylogenies based on mitochondrial data alone have no value. But you should clearly discuss their limits. I also think that your data are worth to be publishes. But they are not complete enough for a good phylogeny of all Lacertini. Add more taxa or restrict your conclusions to those groups of which you have enough samples.

- In the revised version of the manuscript, we restrict ourselves to the phylogeny of Darevskia and included only few critical points about the outgroups, not pretending on the reconstruction of phylogeny of true lizards.

Please look carefully at all the points raised by both reviewers and re-write your manuscript completely. Discussion and conclusions should then be self-critical and show which conclusions are supported by your data and which not. I am looking forward to receive a new manuscript. 

- The MS is rewritten almost completely following the reviewers comments, and some comments are discussed below.

5. Review Comments to the Author

Reviewer #1: The manuscript entitled PHYLOGENY OF CAUCASIAN ROCK LIZARDS (DAREVSKIA) AND OTHER TRUE LIZARDS BASED ON MITOGENOME ANALYSIS: RAPID RADIATION AND ALGORITHM DEPENDENT POSITION OF BASAL NODES is nice contribution presenting whole mitogenome based phylogeny of Darevskia and related genera of true lizards. Authors have sequenced and assembled 16 new mitochondrial genomes of Darevskia species, thus making possible to build phylogeny of Lacertini genera and resolve relationships inside Darevskia genus. The major finding in addition to a sequenced new mitochondrial genomes are detailed and well-grounded workflow of analysis of mitochondrial sequences including approach to minimize influence of substitution saturation. Unfortunately, these real advantages are not properly covered in the discussion and conclusions sections and instead too much space is devoted to discussion of topics which deserves to be a subject of separate research and instead has limited support from this particular study results, therefore the manuscript needs a major revision.

-the authors tried to shift the emphasis in the discussion substantially, in particular they reduce the parts not directly linked with the findings.

Authors did a good job comparing different approaches to whole mitogenome-based phylogeny reconstruction. One of the major findings is that saturation is the main obstacle in reconstruction of robust phylogeny and this could be overcome by careful selection of genes and outgroup or running analyses without outgroup at all. Unfortunately, this have not been reflected in the abstract.

-we tried to discuss this in more detail in the revision, and shift the focus away from the discussion about geological past etc. (only retained in part that is directly related to the genus Darevskia).

In addition, convergence of topologies obtained with different methods could be a good measure of robustness of phylogenetic reconstruction – authors should say it directly. 

-done (Abstract & subsection 4.3)

Finally, I don’t see clear evidence of rapid radiation, just mitochodrial genome comes close to the limit of unambiguously reconstructing of ancient splits.

-agree; the discussion on the potential rapid radiation removed as unnecessary. 

Introduction could be strengthen by adding information about possible flaws of mtDNA in phylogenetic reconstruction, namely only maternal inheritance, capture or lost by lineages via introgression, not-neutrality, functional ties with nuclear genes, possible presence of pseudogenes etc., and how at least some of these difficulties can be targeted when using complete mitochondrial genomes. Please, check these papers

Podnar, M., Haring, E., Pinsker, W., Mayer., W. 2007. Unusual origin of a nuclear pseudogene in the Italian wall lizard: Intergenomic and interspecific transfer of a large section of the mitochondrial genome in the genus Podarcis (Lacertidae). J. Mol. Evol. 64, 308-320.

Godinho, R., Crespo, E.G., Ferrand, M., 2008. The limits of mtDNA phylogeography: complex patterns of population history in a highly structured Iberian lizard are only revealed by the use of nuclear markers. Molecular Ecology. 17, 4670–4683.

https://www.pnas.org/content/99/25/16122?cited-by=yesl99%2F25%2F16122&utm_source=TrendMD&utm_medium=cpc&utm_campaign=Proc_Natl_Acad_Sci_U_S_A_TrendMD_0

https://www.biorxiv.org/content/10.1101/588756v2.abstract

-these issues are addressed in the revised introduction

I think the manuscript should be substantially revised to be more concise, and keep track on the main ideas whose are supported by results. This includes changing title of the paper to more appropriate (rapid radiation is not the only possible reason for absence of strong support of basal nodes; perhaps loosing of phylogenetic signal and non-monotonous rates of molecular evolution among species could also contribute).

- agreed, the title is changed, and the relevant issues addressed in the revision

Other remarks to the text (unfortunately, lines numbering in PDF was missing):

-lines numbering inserted in the revision

“inter-generic phylogenies.” - there is extra return in Abstract

- re-worded

“but BI and ML topologies support basal position of D. parvula from the western Lesser Caucasus.” - did you mean BI and MP?

-of course, thank you for the correction. 

“All topologies shifted the phylogenetic position” - did you mean altered, shuffled?

- altered; corrected

In Methods “The sum of the conflicting, coinciding, and unsupported nodes was considered a measure of similarity between the compared trees” - haven't you consider plotting some of these numbers against saturation indicator to show it’s influence?

- there is not correlation across all studied genes; however 16 S that has the highest valie of the saturation indicator also has the lowest number of conflicting/ unresolved nodes; and ATP8, with the lowest saturation indicator, has the most discrepancies from the mitogenome-based topology, this is explained in the current revision. We retained Table 1 but further simplified it. 

The problem of reconstruction of phylogeny using saturated mitochondrial genes seems not new – see for example https://www.ncbi.nlm.nih.gov/pubmed/11975355
https://www.ncbi.nlm.nih.gov/pubmed/29669515 and it should be discussed

- we mentioned the issue in the introduction.

Fig. 2 legend “Posterior probabilities left of the nodes” does not seem grammatically correct

-corrected

“However, genus-level topologies were still different (results not shown).” - you have Supplementary files where you can show these results.

-we don’t discuss genus-level topologies in the revision, with the exception of two important patterns: basal Takydromus and sister status of Zootoka and Podarcis.

There is subsection “Differences from earlier findings” in Discussion. I think it should be shortened – obviously lots of reconstructions had been made, there is no need to discuss differences in such detailed way.

- the discussion is reduced substantially, according to this recommendation. Besides, we minimized the statements on the Lacertini phylogeny, considering the recommendation of the 2nd referee and the editor, although retained few points that we think are important and well-supported. 

I am surprised not to see molecular dating reconstruction. Authors definitely have very good dataset for that, moreover, they are studying one taxonomic group, which is very much the group of their expertise – why not? It would add value to the publication and justify discussion of biogeographic scenario if it will persist in the final version.

-we prefer to be careful about the dating, since there is very broad range of calibrating molecular clock for lacertids by different authors. However, following the suggestion of the referee, we used two alternative datings, more and less conservative, referring on two important geologicasl events shaping the region: middle Miocene climate transition and Messinian salinity crisis. The respective datings are discussed and illustrated. 

“In the Western Lesser Caucasus area, D. parvula coexists with the representatives of the

next basal clade of Darevskia: the clade ‘D. rudis’ (in sense of (1). Consequently, this area is the

most likely ancestral for all Darevskia. In the original time of divergence, this was the north-east-

ern part of the Anatolian landmass, and the separation could have happened between the water-

sheds separating the upper reaches of the Kura (Mtkvari), Chorokhi (Coruh), and Euphrates Rivers,

due to the growth of xeric and less suitable areas between the lizard habitats.” - Authors may use more formal approach to the ancestral range calculation. Currently, this discussion looks speculative, especially the part with discussion of climate of watersheds. 20 Mya the area could look different due to massive transgressions-regression cycles there – see Popov, Rögl, Rozanov_2004_Lithological-Paleogeographic Maps of Paratethys Cour. Forsch.-Inst. Senckenberg, 2004. — 250 p. Authors definitely must to do additional analyses inferring split time, ancestral areas, ecological features of species of Darevskia genus (ground dwelling versus rock dwelling) and rewrite correspondent sections of manuscript or omit these topics. Actually I would suggest to concentrate here only on molecular dating, since the last two issues would benefit from additional taxa, which are not included in this study.

-we followed this suggestion working on the revision. However, we refrain from discussing ground-vs rock-dwelling species, because the information on likely isolation reasons are insufficient to explain this interesting adaptive trend.

Conclusions must be rewritten according to the new version of manuscript. As an example only, currently “The first is that constructing phylogenies based on the analysis of high volumes of various genetic data is not the best approach for reconstructing species-level phylogenies.” is highly ambiguous;

-Conclusions section re-written and strongly reduced

 “Phylogenetic studies devoted to a specific taxonomic group, such as genera or subfamily, are more trustful for resolving species-level taxonomies than meta-analyses comprising large number of taxa with different evolutionary distances from each other.” is trivial and too general and so on. 

-agree, changed

The last part of last sentence “however some elements of topology are unlikely to be finally resolved even if the complete mitochondrial genome is studied, and, if high-throughput sequencing is not available, it is more useful to combine sequences of mitochondrial and nuclear genes, as (66) recommend.” is merely speculative and is not conclusion of this research.

-removed

Reviewer #2: In the present manuscript the authors sequence 16 mitochondrial genomes of the lacertid genus Darevskia and use them to reach conclusions of the phylogenetic relationships within this genus and, together with GenBank sequences, to infer the phylogenetic relationships within Lacertini. Although sequencing 16 mitochondrial genomes is a good effort it is certainly not the way forward to solve evolutionary relationships in the XXI century. The mtDNA is very long an can provide quite a lot of information but the main drawback is that it represents a single locus. In fact, mtDNA is a very particular locus with exclusively maternal inheritance and with a high evolutionary rate. Aspects that have proven very good for resolving many evolutionary relationships but that have also been shown to recover wrong phylogenetic relationships in many occasions. For this reason, in the last 10 years it has been very usual to apply a multilocus approach to phylogenetic inference, first with a few loci (2-3) but latter on with more (5-8) and more recently with hundreds or even thousands of loci. Of course, missing data is still an issue but multilocus phylogenies are certainly the way forward to obtaining the correct evolutionary relationships of taxa. Especially on these groups in which the phylogenetic relstionships are contentious. NGS techniques have opened new doors to dig further into the phylogenetic relationships of difficult groups (including Lacertini) using many independent loci. To suddenly apply these NGS techniques to obtain long stretches of a highly variable and maternally inherited single locus is like going back to the beginning of phylogenetic inference. In fact, one of the conclusions of the present manuscript is that just to genes, 16S and cytb, give the same (or very similar) results than the whole mitogenomes. Phylogenies of 16S and cytb have been around for more than 20 years and luckily technical and analytical advances occurred in the last 10 years have allowed a leap forward towards multilocus phylogenies, a much better approach than single locus phylogenies like the one presented in the present study.

- We know that it is possible to sequence nuclear genes and also use genomic data. However, (1) multilocus approach has its own shortages for reconstruction of low-level phylogenies, where gene introgression commonly takes place, or incomplete lineage sorting may be the case. In fact, mitochondrial DNA phylogeny may not completely coincide with species-level phylogeny but its interpretation is straightforward – this is phylogeny of maternal lineages. This is indicated in the text. (2) as mentioned in the paper, there are hundreds of DNA laboratories worldwide that are and will be during quite long time rely on PCR based methods and individual gene sequences. If answers on some specific questions can be found with less equipment and cheaper techniques, we should not reject this opportunity. 

In the manuscript the authors use a largely incomplete phylogeny of Lacertidae to discuss the phylogenetic relationships within Lacertini. The results presented are completely uninteresting and definitively not worth publishing. The authors should delete this whole section from the paper. There have been recent studies using multilocus data that have not been included in the manuscript (Mendes et al. 2016 Molecular Phylogenetics and Evolution 100 (2016) 254–267) and this makes the whole discussion comparing to papers by Fu et al. and Harris et al from 1997 and 1998 and other older papers irrelevant and uninteresting. The authors are missing many Lacertini genera and therefore they should not use their incomplete dataset to discuss the phylogenetic relationships within this very relevant group. Doing that is a superfluous and confusing exercise that is, again, not worth publishing. After Mendes et al. 2016 the phylogeny of Lacertini is more or less fixed until new studies using thousands of loci obtained using NGS will be published in 2020. Datasets should be developed to answer specific questions and this partial mitogenomic dataset is definitively not suitable to address the phylogeny of Lacertini.

-the paper of Mendes et al. is cited in the revision. The discussion on the inter-generic relationships of Lacertini is strongly reduced. 

Across the whole manuscript the authors use the word “basal” incorrectly. I urge the authors to read this paper very carefully and correct the way they discuss phylogenies.

https://www.csun.edu/~dgray/Evol322/OmlandTreeThinking.pdf

-The authors are familiar with the basics of cladistics philosophy, and they did not use the word “basal” in sense of “ancestral” which seems to be the concern of this reviewer--Basal is used in sense of the branch or clade in the phylogeny most distant (and earliest separated) from the rest of the evolutionary lineages. We do not agree with the reviewer that this term is used incorrectly throughout the manuscript. 

The exercise within Darevskia is interesting but again it is restricted to half of the described species. In fact, they only include species within Georgia.

- The species set included in the analysis represents all clades and major groups of Darevskia, except group D. defillippi. There are species of the genus found far south, west, and east from the sampling area, however, they are either found in Georgia as well, or their closest relatives are found there. The only group of species not found in Georgia is D. chlorogaster, and the sample of this species collected from SE Azerbaijan is included. Because all of the major clades are well represented, and because relationships of the more terminal taxa are not at issue, we feel that the species presented in this analysis provide an optimal assortment of Darevskia taxa, commensurate with our limited resources and time.

Minor comments:

P12: Neither in the Fig. legend or in the figure itself the authors indicate Georgia. They should indicate where is Georgia.

-corrected

P14: The D. parvula is most probably D. adjarica (previously D. parvula adjarica). See https://biotaxa.org/Zootaxa/article/view/zootaxa.4472.1.3

-according to the Reptile Database (http://www.reptile-database.org/) it is still D. p. adjarica, we rely on this nomenclature.

P14: D. clarcorum is wrongly written in several places in the text (OK in the map). It is d. clarkorum, with “k”.

-corrected

Pag 14: “E. multcocellata” is E. multiocellata

-corrected

Pag 17: in several places they use “polytiomies”, correct to “polytomies”

-corrected

Pag 21: Legend for Fig. 4 “genes” is repeated twice

-corrected

Within Darevskia, “parvula” (or better adjarica; see Arribas et al. 2018) is sister taxon to the remaining Darevskia included in the study. Despite they present it as a new finding, it had been already published by Fu et al. (1997) and mentioned by Arribas et al (2018).

Please, check this paragraph from Arribas et al. (2018):

“Grechko et al. (1997) found that D. parvula was sister taxon to all the other Darevskia when studied using “DNA Taxon prints” (Restriction endonuclease analysis of Highly Repetitive DNA). Ciobanu et al. (2002, 2003) and Grechko et al. (2006) also studied Satellite DNA in Darevskia and found a relatively isolated position of D. parvula in which only one type of Satellite DNA predominate (CLsatIII), and as inhabitants of one of the glacial refuges in the extremes of the Caucasus (as D. clarkorum/dryada) in the west and D. chlorogaster in the southeast, they argue that possibly they resemble the ancestors of the group. The presence of these same microsatellites in the “mixta-group” (clarkorum, caucasica, daghestanica, dryada) casts some doubts on the basal association of D. parvula with the rudis group as stated by means of mtDNA (see above). For Grechko et al. (2006), D. parvula would be outside these two clades, but hybridized in an early instance of its diversification with the ancestors of both clades.”

- Thank you for this comment. Fu et al. (1997) suggested basal position for D. praticola, and not D. parvula. Grechko et al. indeed hypothesized this based on the microsatellite and restriction endonuclease analysis, and we mention this in the revision. 

-At the end of page 17 they comment: “We did not include D. alpina in our analysis, however, this species is matrilineally very close to D. saxicola, and may be of hybrid origin”. What is possibly of hybrid origin is not the species but the individual sewuenced by Murphy et al. To extrapolate it to the whole species is a bit excessive.

- the statement is re-worded. 

Page 27 etc. As mentioned in the general comments above, the manuscript is not suitable for addressing the phylogenetic relationships and/or the biogeography of lacertini. It is missing many taxa, it uses a single locus and therefore it is wrong and confusing. I would suggest the authors to refrain from publishing these results on the phylogeny and biogeography of Lacertini

-as said above, the respective sections are removed or greatly reduced

---

## [Decision Letter · Decision Letter 2]

19 Mar 2020

PONE-D-19-16166R2

PHYLOGENY OF CAUCASIAN ROCK LIZARDS (DAREVSKIA) AND SOME OTHER TRUE LIZARDS BASED ON MITOGENOME ANALYSIS: OPTIMISATION OF THE ALGORITHMS AND GENE SELECTION

PLOS ONE

Dear Dr Tarkhnishvili,

Thank you for submitting your manuscript to PLOS ONE. After careful consideration, we feel that it has merit but does not fully meet PLOS ONE’s publication criteria as it currently stands. Therefore, we invite you to submit a revised version of the manuscript that addresses the points raised during the review process.

We would appreciate receiving your revised manuscript by May 03 2020 11:59PM. To enhance the reproducibility of your results, we recommend that if applicable you deposit your laboratory protocols in protocols.io, where a protocol can be assigned its own identifier (DOI) such that it can be cited independently in the future. For instructions see: http://journals.plos.org/plosone/s/submission-guidelines#loc-laboratory-protocols

We look forward to receiving your revised manuscript.

Kind regards,

Ulrich Joger

Academic Editor

PLOS ONE

Additional Editor Comments (if provided):

Please obey the requests by reviewer 2 and add some explanation of the tree.

Reviewers' comments:

Reviewer's Responses to Questions

**Comments to the Author**

1. If the authors have adequately addressed your comments raised in a previous round of review and you feel that this manuscript is now acceptable for publication, you may indicate that here to bypass the “Comments to the Author” section, enter your conflict of interest statement in the “Confidential to Editor” section, and submit your "Accept" recommendation.

Reviewer #1: (No Response)

Reviewer #2: All comments have been addressed

2. Is the manuscript technically sound, and do the data support the conclusions?

Reviewer #1: Yes

Reviewer #2: Yes

3. Has the statistical analysis been performed appropriately and rigorously? 

Reviewer #1: Yes

Reviewer #2: Yes

4. Have the authors made all data underlying the findings in their manuscript fully available?

Reviewer #1: Yes

Reviewer #2: Yes

5. Is the manuscript presented in an intelligible fashion and written in standard English?

Reviewer #1: Yes

Reviewer #2: Yes

6. Review Comments to the Author

Reviewer #1: Line 196 “were:Podarcis” - space missing

L. 203 “E. multcocellata” – again!

Line 214 “);,” - delete semicolon

Line 269 “D. Clarkorum” - unnecessary capitalization

Line 474 “odd Castoe” sounds odd

Sorry to say, but what I am still missing is the description of methods and results of time-calibrated tree. The Figure 6 pops up from nowhere first time already in Discussion section. Please, add few lines how this analysis was done and what is showed in terms of topology of tree, convergence of parameters, dates etc. I believe, that this is my last request.

Reviewer #2: As mention in my previous reviews, I do not like the present manuscript. I think that by being mtDNA only it is more misleading than helpful; even if what is use are mtDNA genomes. Said that, the authors have done an effort to include most of the comments by the referees. They still are wrong with the use of the term "basal" but as I said in my previous reviews with this and many other errors that they had: it is not my paper and therefore it is up to them to fix it. Please have a look to the following paper:

https://onlinelibrary.wiley.com/doi/full/10.1111/j.0307-6970.2004.00262.x

7. PLOS authors have the option to publish the peer review history of their article (what does this mean?). If published, this will include your full peer review and any attached files.

Reviewer #1: No

Reviewer #2: No

---

## [Author Response · Author response to Decision Letter 2]

25 Mar 2020

Additional Editor Comments (if provided):

Please obey the requests by reviewer 2 and add some explanation of the tree.

- the request of the reviewer 2 considering the terminology is considered, and the time-tree is additionally explained in the methods subsection 2.4.

Reviewer #1: Line 196 “were:Podarcis” - space missing

- 

- corrected

L. 203 “E. multcocellata” – again!

- Of course, multiocellata. Corrected.

Line 214 “);,” - delete semicolon

- corrected

Line 269 “D. Clarkorum” - unnecessary capitalization

- corrected

Line 474 “odd Castoe” sounds odd

- of course, Todd Castoe. Corrected.

Sorry to say, but what I am still missing is the description of methods and results of time-calibrated tree. The Figure 6 pops up from nowhere first time already in Discussion section. Please, add few lines how this analysis was done and what is showed in terms of topology of tree, convergence of parameters, dates etc. I believe, that this is my last request.

- For inferring time-calibrated tree we used the same BEAST runs as for reconstructing full-mitogenome-based BI phylogeny, using node calibration suggested by Garcia-Paris et al. We added few lines to the Methods subsection 2.4 to explain this.

Reviewer #2: As mention in my previous reviews, I do not like the present manuscript. I think that by being mtDNA only it is more misleading than helpful; even if what is use are mtDNA genomes. Said that, the authors have done an effort to include most of the comments by the referees. They still are wrong with the use of the term "basal" but as I said in my previous reviews with this and many other errors that they had: it is not my paper and therefore it is up to them to fix it. Please have a look to the following paper:

https://onlinelibrary.wiley.com/doi/full/10.1111/j.0307-6970.2004.00262.x

- The authors discussed the use of the term “basal” and decided to follow the recommendation of the reviewer. In the revised version, we are speaking about “basal branchings” rather than basal taxa.

---

## [Decision Letter · Decision Letter 3]

12 May 2020

PHYLOGENY OF CAUCASIAN ROCK LIZARDS (DAREVSKIA) AND SOME OTHER TRUE LIZARDS BASED ON MITOGENOME ANALYSIS: OPTIMISATION OF THE ALGORITHMS AND GENE SELECTION

PONE-D-19-16166R3

Dear Dr. Tarkhnishvili,

We are pleased to inform you that your manuscript has been judged scientifically suitable for publication and will be formally accepted for publication once it complies with all outstanding technical requirements.

With kind regards,

Ulrich Joger

Academic Editor

PLOS ONE

Additional Editor Comments (optional):

Reviewers' comments:

Reviewer's Responses to Questions

**Comments to the Author**

1. If the authors have adequately addressed your comments raised in a previous round of review and you feel that this manuscript is now acceptable for publication, you may indicate that here to bypass the “Comments to the Author” section, enter your conflict of interest statement in the “Confidential to Editor” section, and submit your "Accept" recommendation.

Reviewer #1: All comments have been addressed

2. Is the manuscript technically sound, and do the data support the conclusions?

Reviewer #1: Yes

3. Has the statistical analysis been performed appropriately and rigorously? 

Reviewer #1: Yes

4. Have the authors made all data underlying the findings in their manuscript fully available?

Reviewer #1: Yes

5. Is the manuscript presented in an intelligible fashion and written in standard English?

Reviewer #1: Yes

6. Review Comments to the Author

Reviewer #1: I have no further suggestions or corrections for the manuscript.

in line 236 “Lacertii” should be Lacertini

7. PLOS authors have the option to publish the peer review history of their article (what does this mean?). If published, this will include your full peer review and any attached files.

Reviewer #1: No

---

## [Editor Report · Acceptance letter]

14 May 2020

PONE-D-19-16166R3 

Phylogeny of Caucasian Rock Lizards (*Darevskia*) and Other True Lizards Based on Mitogenome Analysis: Optimisation of the Algorithms and Gene Selection 

Dear Dr. Tarkhnishvili:

I am pleased to inform you that your manuscript has been deemed suitable for publication in PLOS ONE. Congratulations! Your manuscript is now with our production department. 

With kind regards,

on behalf of

Dr. Ulrich Joger 

Academic Editor

PLOS ONE